# Predictive features of gene expression variation reveal mechanistic link with differential expression

Olga M Sigalova[1] [iD], Amirreza Shaeiri[2] [iD], Mattia Forneris[1] [iD], Eileen EM Furlong[1],* [iD] & Judith B Zaugg[2],** [iD]

## Abstract

For most biological processes, organisms must respond to extrinsic cues, while maintaining essential gene expression programmes. Although studied extensively in single cells, it is still unclear how variation is controlled in multicellular organisms. Here, we used a machine-learning approach to identify genomic features that are predictive of genes with high versus low variation in their expression across individuals, using bulk data to remove stochastic cell-to-cell variation. Using embryonic gene expression across 75 *Drosophila* isogenic lines, we identify features predictive of expression variation (controlling for expression level), many of which are promoter-related. Genes with low variation fall into two classes reflecting different mechanisms to maintain robust expression, while genes with high variation seem to lack both types of stabilizing mechanisms. Applying this framework to humans revealed similar predictive features, indicating that promoter architecture is an ancient mechanism to control expression variation. Remarkably, expression variation features could also partially predict differential expression after diverse perturbations in both *Drosophila* and humans. Differential gene expression signatures may therefore be partially explained by genetically encoded gene-specific features, unrelated to the studied treatment.

**Keywords** embryogenesis; expression variation; gene expression; promoters; transcriptional regulation
**Subject Categories** Chromatin, Transcription & Genomics; Computational Biology
**Mol Syst Biol. (2020) 16: e9539**

## Introduction

Living systems have a remarkable capacity to give rise to robust and highly reproducible phenotypes. Perhaps the most striking example of this is the process of embryogenesis, where fertilized eggs give rise to stereotypic body plans despite segregating genetic variants and moderate differences in environmental conditions (e.g. water temperature for fish and mothers' diet for humans). This phenomenon led Waddington to propose that developmental reactions are canalized, and buffered to withstand such variation without major alterations in embryonic development (Waddington, 1942). In agreement with this, variation in gene expression is an evolvable trait under selection pressure (Fraser *et al*, 2004; Lehner, 2008; Metzger *et al*, 2015).

Gene expression variation can arise from a multitude of stochastic, environmental and genetic factors (Raser & O'Shea, 2005; Huang, 2009; Félix & Barkoulas, 2015; Eling *et al*, 2019). For some genes, expression variation is tolerated, without obvious effects on fitness, or can even be beneficial, for example in stress response or in stochastic cell fate decisions (Blake *et al*, 2006; Raj & van Oudenaarden, 2008; Macneil & Walhout, 2011). In other cases, variation in gene expression is detrimental and must be tightly regulated, for example for essential genes (Fraser *et al*, 2004) and genes that reduce fitness in heterozygous mutants (Batada & Hurst, 2007). This suggests that there are inherent mechanisms that modulate variation in gene expression, either attenuating or amplifying it (Fig 1A).

Studies on single-celled organisms or in cell lines have linked multiple regulatory mechanisms to gene expression variation, including the presence of a TATA-box at the gene's promoter (Blake *et al*, 2006; Ravarani *et al*, 2016), CpG islands (Morgan & Marioni, 2018), bivalent chromatin marks (Faure *et al*, 2017), polymerase pausing (Boettiger & Levine, 2009) or miRNA binding (Schmiedel *et al*, 2015). However, it remains unclear what mechanisms regulate expression variation in multicellular, developing organisms in a gene- and tissue-specific manner.

To address this, we devised a machine-learning approach and performed a systematic analysis of factors underlying variation in gene expression during embryogenesis to uncover the regulatory mechanisms involved. To measure expression variation, we used gene expression data generated from a pool of embryos (~100) sampled from 75 different isogenic *Drosophila melanogaster* lines during embryogenesis (Cannavò *et al*, 2017). This experimental design cancels out most stochastic noise (as it is bulk sequencing), tissue-specific expression pattern (as it is whole embryo) and slight differences in developmental progression (as it is 100 embryos per

1 Genome Biology Unit, European Molecular Biology Laboratory (EMBL), Heidelberg, Germany
2 Structures and Computational Biology Unit, European Molecular Biology Laboratory (EMBL), Heidelberg, Germany
*Corresponding author. Tel: +49 6221 387 8416; E-mail: furlong@embl.de
**Corresponding author. Tel: +49 6221 387 8528; E-mail: judith.zaugg@embl.de

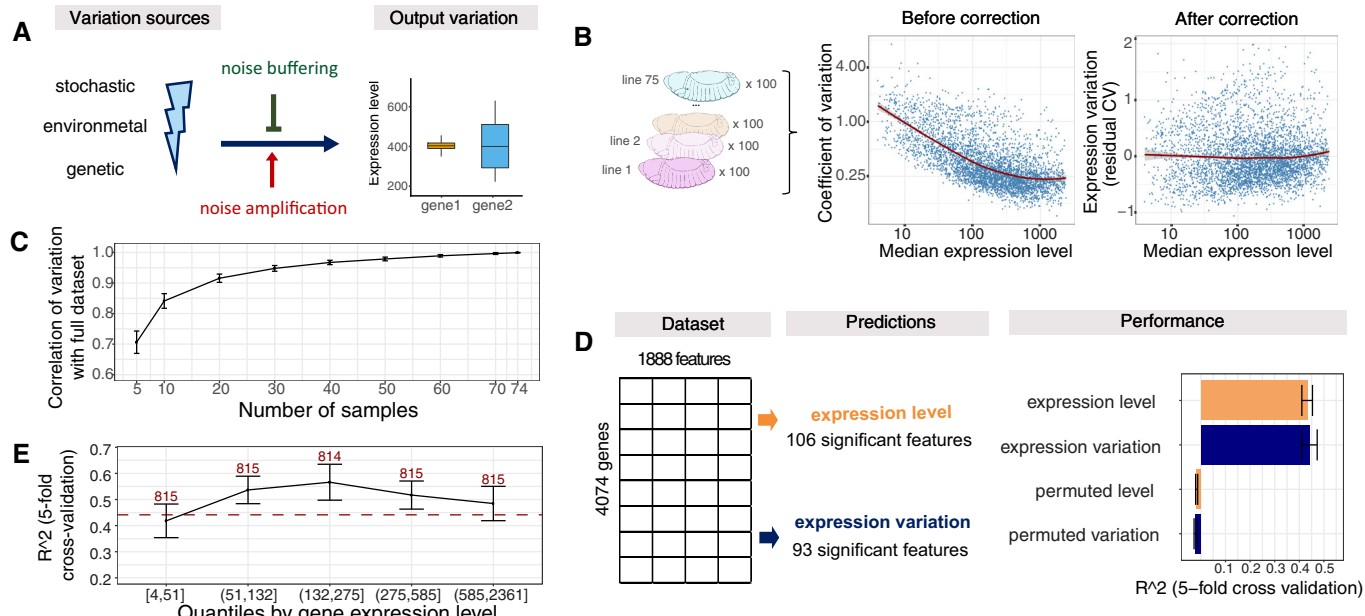

**Figure 1. Genomic features can predict expression variation independent of expression levels.**

A  Differences of gene regulatory mechanisms related to noise amplification and noise buffering would result in different observed expression variation given the same variation sources (left).

B  Dependence between coefficient of variation (CV) and median expression level of 4,074 genes across 75 samples (left). Residuals from LOESS regression of CV on the median were used as the measure of variation throughout the analysis (right). Median expression level and coefficient of variation plotted on log2 scale, red line represents LOESS regression fit.

C  Correlation of expression variation calculated from subsets of samples versus the full dataset (Pearson correlation coefficient). Data are presented as mean ± SD (100 independent selections of samples).

D  Schematic overview of the random forest models and feature selection with the Boruta algorithm (left). Performance shown as $R^2$ from fivefold cross-validation and compared to randomly permuted data (right). Data are presented as mean ± SD (fivefold cross-validation).

E  Performance ($R^2$, fivefold cross-validation) for genes grouped by expression levels (quantiles). Data are presented as mean ± SD (fivefold cross-validation), number of genes per quantile indicated (x-axis). Red dotted line indicates performance of full model.

sample). To dissect the regulatory mechanisms that modulate expression variation (Fig 1A), we collated over a thousand gene-specific and genomically encoded features and applied a random forest model to identify the properties that best explain expression variation across individuals. As a comparison, we also predict median expression level across lines using the same features.

Our results show that, overall, increasing regulatory complexity translates into more robust gene expression. We identified two independent mechanisms associated with genes with low expression variation across individuals: low variable genes either have (i) a broad transcription initiation region (broad promoters) with high transcription factor (TF) occupancy, or (ii) narrow initiation regions (narrow promoters) with RNA Polymerase II (Pol II) pausing and high regulatory complexity outside their promoter region. In contrast, genes with high variability generally have narrow promoters and little other regulatory features, suggesting that it may be rather a lack of "stabilizing" mechanisms that facilitate their noisy expression. Applying the same framework to human data derived from tissues across individuals (Lonsdale *et al*, 2013) identified similar promoter-associated features to be predictive of expression variation, thus validating our findings in an independent organism. Remarkably, these same features are also predictive of differentially expressed genes when tested on independent datasets from adult *Drosophila* subjected to different stress conditions and genetic

perturbations, and in a collection of differential expression data for humans. These findings suggest that the differential expression response may be partially explained by genetically encoded gene-specific features that are unrelated to the treatment applied.

Taken together, our results suggest that gene expression variation across genetically diverse multicellular organisms is strongly linked to how the gene is regulated and may reflect evolutionary constraints on expression precision.

## Results

### Measuring gene expression variation across individuals

To understand the mechanisms by which gene expression variation is controlled during embryonic development, we obtained RNA-seq data from 75 isogenic lines of *Drosophila melanogaster* embryos at three different developmental stages (2–4, 6–8 and 10–12 h post-fertilization) from Cannavò *et al* (2017). To reduce potential confounding effects of maternally deposited RNA, we focused on the late embryonic time-point (10–12 h after fertilization) and removed genes whose expression decreased between 2–4 and 10–12 h, resulting in embryonic expression data for 4,074 genes (Materials and Methods, Appendix Fig S1). For each gene, we calculated

its median expression level and the coefficient of variation (CV) from the normalized read counts across individuals (Materials and Methods). As variation is highly correlated with the levels of gene expression (Anders & Huber, 2010; Ran & Daye, 2017; Eling *et al*, 2018), we used the residuals from a locally weighted regression (LOESS) of the CV on median expression to obtain a measure of expression variation that is relative to the expected variation at a given expression level (Fig 1B).

We confirmed that this measure of variation is highly correlated with alternative metrics, such as variance-stabilized standard deviation or residual median absolute deviation (Appendix Fig S2A and B), and robust with respect to the number and identity of samples used (Fig 1C). Moreover, using the full dataset from Cannavò (Cannavò *et al*, 2017), expression variation values were highly correlated across time, especially for consecutive time-points, further confirming the approach (Materials and Methods, Appendix Fig S1D). Finally, we observed a strong correlation in expression variation between pairs of genes in close proximity (Appendix Fig S1E), as previously observed for neighbouring genes in yeast (Becskei *et al*, 2005; Batada & Hurst, 2007).

As these 75 samples came from strains with different genotypes, we first calculated the proportion of expression variance that is explained by genetics in *cis* (taking variants within 50 kb of each gene into account) using variance decomposition (Materials and Methods). On average, 6% (median across all genes) of the total gene expression variation was explained by *cis* genetics (Appendix Fig S1F), indicating that more complex genetic effects and other properties must account for the majority of expression variation. We reasoned that differences in the extent of expression variation among genes should reflect inherent differences in their regulation, including their regulatory complexity and mechanisms of noise buffering or amplification. Therefore, in the remainder of this study we investigate the regulatory differences between genes with high versus low expression variation.

## Genomic features predict expression variation independent of expression levels

To understand the drivers of expression variation, we collected 1,888 gene-specific features (Datasets EV1 and EV2) and used random forest regression to identify those that are associated with either expression variation or expression level (Fig 1D). This allowed us to distinguish between features that are predictive of one or both properties. The features can be broadly divided into seven categories: transcription start site (TSS, e.g. core promoter motifs, chromatin accessibility, TF binding), gene body features (e.g. gene length, number of exons), 3′untranslated regions (UTR, e.g. length, miRNA motifs), distal regulatory elements (e.g. TSS-distal chromatin accessibility, TF occupancy), gene type (e.g. housekeeping genes, TFs), gene context (e.g. gene density, distance to the borders of topologically associated domains (TADs)) and genetics (e.g. the presence of eQTL and a *cis* genetic component; full description in Materials and Methods and Dataset EV1).

To restrict our analysis to the important features, we applied the random forest-based Boruta algorithm, which iteratively selects all features that predict better than their permuted version (Kursa & Rudnicki, 2010). This resulted in 93 and 106 predictive features for expression variation and level, respectively (Fig 1D). Using these feature sets, our models predicted expression variation and level with an $R^2$ of 0.45 and 0.43 (fivefold cross-validation), respectively, while permuting the labels resulted an $R^2$ of zero (Fig 1D). Including gene expression as a feature leads to a slight improvement in the random forest performance ($\Delta R^2 = 0.07$, Appendix Fig S2C), even though expression level and variation are globally uncorrelated by construction (Fig 1B). To avoid any confounding effects, we decided to exclude expression level from the features to predict expression variation, and instead report results for predicting expression level and expression variation side by side.

To ensure the robustness of our predictions, we performed a number of analyses: first, we verified that the predictions for variation are independent of the level of gene expression by showing that the models performed equally well on genes grouped into quartiles based on their expression levels (Fig 1E). Second, we ensured that the predictions are robust to the choice of measure used for expression variation (Appendix Fig S2D). Third, we tested whether dynamic gene expression changes during developmental stages can contribute to the variation predictions. We reran the random forest models, predicting expression variation for genes grouped based on their absolute expression change between 6–8 and 10–12 h after fertilization. For genes with minor expression change between the two time-points (below median of 0.8), the performance was comparable to the full model, while for the genes with a stronger expression change (above 0.8), the $R^2$ dropped to about 0.3 (Appendix Fig S2E). This indicates that some portion of expression variation comes from dynamic changes in gene expression during embryogenesis, which is not captured by our features (and thus reduces the performance of our model for this set of genes). However, since the performance is the best for genes that vary little between stages, it indicates that variance explained by our model is overall not majorly confounded by expression dynamics. Finally, the model performance does not decrease when training and test sets come from different chromosomes (Appendix Fig S2F), demonstrating that the results are not confounded by shared regulatory features between neighbouring genes.

Taken together, these results establish that gene expression variation—as well as gene expression levels—can be predicted based on genomically encoded features, when measured across a population of genetically diverse individuals during embryogenesis. The predictions are independent of the gene's expression level and are robust to the metric used for measuring variation. These models can therefore be used as the basis for addressing questions about buffering mechanisms that regulate gene expression variation during embryogenesis.

## Promoter architecture is the most important predictor of expression variation

Next, we used this predictive framework to investigate the genomic features that best explain expression variation and expression level. We retrieved the features' "importance score" from the Boruta algorithm and determined the correlation of each feature with both expression properties (Dataset EV4). Although most features are to some extent predictive of both expression level and variation, their relative importance differed substantially (Fig 2A). Being a housekeeping gene, for example, was strongly predictive of high expression level while being less important for expression variation.

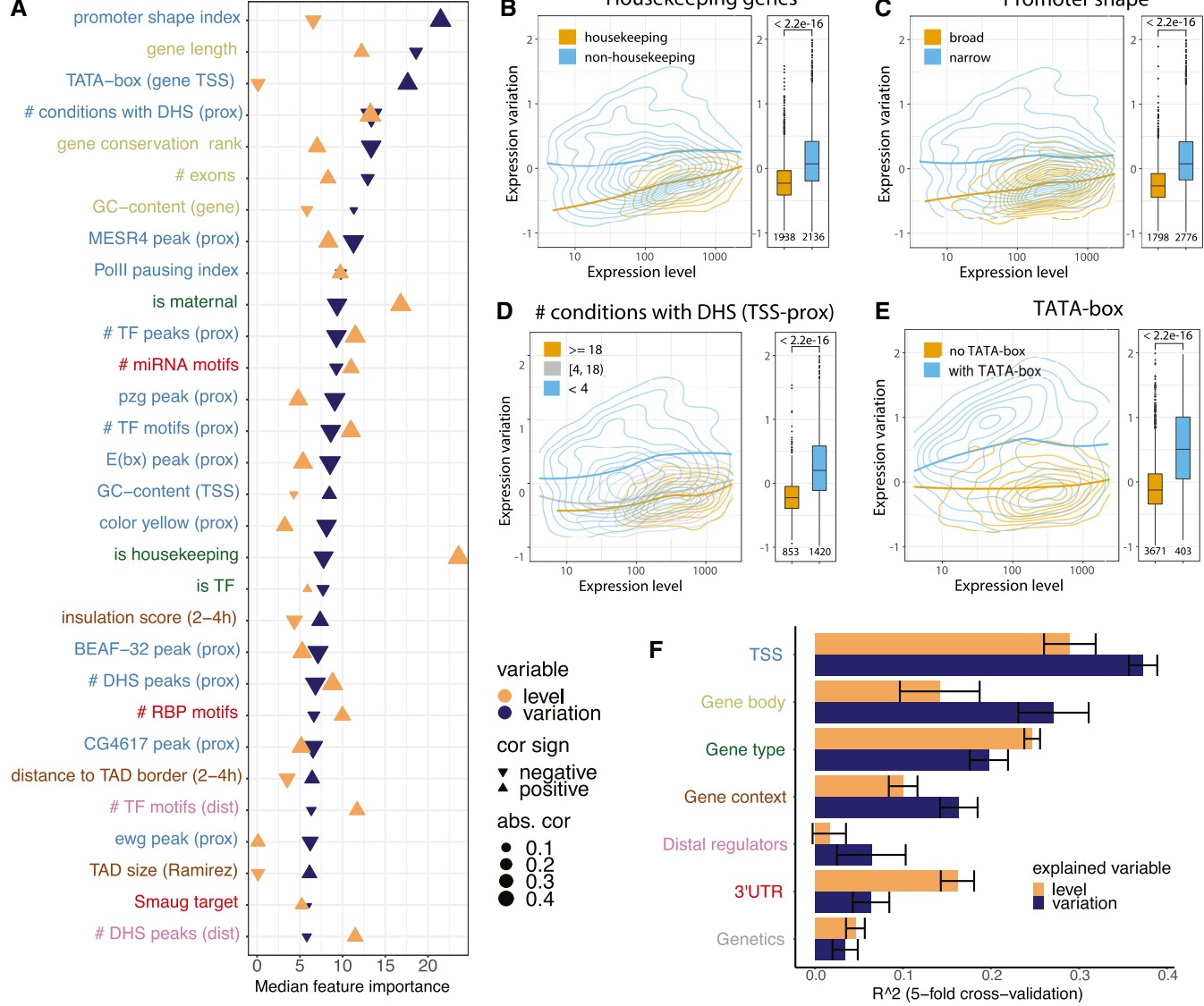

Figure 2. Promoter architecture is the most important predictor of expression variation.

A Top 30 important features for predicting expression variation using the Boruta feature selection. Features are ordered by their importance for expression variation (blue) and show the corresponding importance for level (orange). The absolute value and sign of correlation coefficient are indicated by the triangle size and orientation, respectively. For binary features, phi coefficient of correlation was used, otherwise Spearman's coefficient of correlation. Label colours correspond to feature groups in (F).

B–E Relationship between expression level and expression variation shown as 2D kernel density contours (left) and boxplots (right) for housekeeping genes (B, Cohen's d = 0.8), genes separated by promoter shape (C, Cohen's d = 1.0), number of embryonic conditions with a TSS-proximal DHS (D, Cohen's d = 1.1 for top vs. bottom group) and presence of TATA-box at TSS (E, Cohen's d = 1.4). LOESS regression lines indicated for each gene group, P-values from Wilcoxon test. Boxplots: median as central band, the first and the third quartiles as lower and upper hinges, the upper and lower whiskers extend from the hinge to the largest/smallest value at most 1.5 inter-quartile range, respectively. Numbers of genes in each group are shown under the boxplots.

F Performance of random forest predictions ($R^2$) for expression level (orange) and variation (blue) trained on individual feature groups. Data are presented as mean ± SD (fivefold cross-validation), colour code of y-axis labels matches Fig 2A.

Conversely, the presence of a core promoter TATA-box motif is strongly predictive of high expression variation only (Fig 2A; see Dataset EV4 for full list). We note that most features are either associated with higher variation and lower expression or vice versa, suggesting that expression level and variation are not completely independent, as was previously observed (Faure et al, 2017), even though they are globally uncorrelated (Fig 1B). However, we found

that when we split genes into the categories of the top features (e.g. housekeeping vs. non-housekeeping) the differences in expression variation are pronounced at all expression levels (Fig 2B–E): for example, housekeeping genes (the strongest predictor of expression level) are less variable than non-housekeeping genes at any level of expression (Fig 2B). The same holds true for the "Promoter shape" feature, which is the strongest predictor of variation (Fig 2C), as

well as other features such as number of developmental conditions in which a gene had a TSS-proximal DNase hypersensitive site ("# conditions with DHS (prox)"; Fig 2D) and the presence of a TATA-box at TSS ("TATA-box"; Fig 2E). This demonstrates that the features explain expression variation independent of expression level.

Promoter-associated features (*TSS-proximal*) are among the strongest predictors in terms of explanatory power for expression variation, and include promoter shape, core promoter motifs and GC-content, Pol II pausing, chromatin accessibility, and TF occupancy at TSS (Fig 2A). Consequently, a model based only on TSS-proximal features can predict expression variation fairly well with $R^2 = 0.37$ while performing less well for predicting expression level ($R^2 = 0.29$; Fig 2F). Although lower than the model using all features ($R^2$ of variation/level 0.45/0.43), this is markedly higher than a model on any other feature type alone. For example, the next most predictive feature classes for variation are *gene body* ($R^2$ 0.27/ 0.14) and *gene type* (0.20/0.25; although more predictive of expression level), followed by *gene context* (0.16/0.10). *3′UTR* features, which rank third among the most predictive features of expression levels, show little predictive value for variation (0.06/0.16), and distal features overall showed a rather weak predictive value for both variation and level (0.06/0.01). Finally, *Genetics* was the least predictive for both variation and level among the seven feature groups (0.03/0.05), in keeping with the variance decomposition analysis above.

In summary, our results demonstrate that multiple regulatory features can independently predict gene expression variation or gene expression levels. Interestingly, promoter features, rather than upstream regulatory complexity (such as distal DHS sites), are the most predictive of expression variation. This is most likely due to inherent properties of the promoter itself, as promoter architecture is associated with variation in transcriptional start site usage across individuals (Schor *et al*, 2017). Given that housekeeping genes and TFs tend to have different promoter types (Lenhard *et al*, 2012; Arnold *et al*, 2017; Haberle & Stark, 2018), this suggests that specific biological functions may have distinct mechanisms to reduce variation and provide robustness to their expression (e.g. broad promoters; Schor *et al* (2017)) as seen by models based solely on a gene's functional annotation (Gene type in Fig 2F). We cannot exclude that the low predictive power of some features may partially result from incomplete information, such as a lack of high-confidence genome-wide enhancer–gene associations, or masking of *cis* genetic variation by *trans*-acting factors.

## Expression variation in broad versus narrow promoter genes reflects trade-off between expression robustness and plasticity

The most prominent predictive feature for expression variation is promoter shape index (Fig 2A), which classifies promoters based on the broadness of their transcriptional initiation region (Rach *et al*, 2009; Lenhard *et al*, 2012; FANTOM Consortium and the RIKEN PMI and CLST (DGT) *et al*, 2014; Schor *et al*, 2017). Genes with narrow promoters generally have higher variation compared to genes with broad promoters (Fig 2C), and, interestingly, also comprise a wider range of variation (Fig 3A). Moreover, expression variation of narrow promoter genes is better explained by genomically encoded features compared to broad promoter genes

($R^2 = 0.37$ vs. 0.14), and this difference in performance becomes more pronounced with more stringently defined narrow and broad promoter genes (Fig 3B). In contrast, broad promoters themselves are more robust, their genes have generally less variation in their expression (Fig 3A and B), and their promoters are more tolerate of the presence of genetic variants (Schor *et al*, 2017).

Interestingly, when we group genes from the two promoter classes into quartiles based on their variation, we find very specific functions enriched among them: the broad class is strongly enriched for housekeeping genes (Fisher's test odds ratio, OR = 15.0, *P*-value < 1e-16, Dataset EV5) and GO terms related to basic cellular processes (cellular transport, secretion and DNA/RNA biogenesis), with the exception of the top 25% of the most variable genes within the group being strongly enriched in housekeeping metabolic processes (Fig 3C, Appendix Fig S3A, Dataset EV6). In contrast, narrow promoter genes fall into two functional categories depending on their expression variation: the bottom 50% were enriched in TFs (OR = 3.0, *P*-value < 1e-16) and GO terms related to development, signalling and regulation of transcription, while the top 50% are enriched in TATA-box genes (OR = 7.9, *P*-value < 1e-16) and GO terms related to metabolism, stress response and cuticle development (Fig 3C, Appendix Fig S3A). We therefore grouped genes along the dimensions of promoter shape and expression variation into three classes (Fig 3A): genes with broad promoters and low levels of variation in expression (broad), genes with narrow promoters and low expression variation (narrow-low) and genes with narrow promoters and high expression variation (narrow-high).

Next, we looked at regulatory plasticity of these classes of genes, defined here as the variation in accessibility (DHS signal) at the promoter across tissues and developmental time (Materials and Methods). We observed that narrow promoter genes had high regulatory plasticity regardless of their expression variation (Appendix Fig S3B). In particular, narrow-low genes are robustly expressed across individuals at the given developmental stage while having condition-specific regulation. In contrast, broad promoter genes are characterized by both low expression variation and low plasticity, which agrees with their housekeeping functions.

Enrichment of low-variable genes in either housekeeping (*broad*) or developmental (*narrow-low*) functions may reflect selection pressure to reduce expression noise in genes essential for viability and development, as suggested by previous studies (Fraser *et al*, 2004; Lehner, 2008; Metzger *et al*, 2015). One proxy for evolutionary constraints is sequence conservation across long evolutionary distances. In keeping with this, sequence conservation between *Drosophila* and human was among the top five most predictive features of low expression variation, with conserved genes being significantly less variable (Fig 2A, Wilcoxon test, *P*-value < 2e-16). Promoter shape is also correlated with gene conservation: conserved genes are highly enriched for broad promoters (80% in broad vs. 41% in narrow) and more enriched in the narrow-low compared to narrow-high class (54 vs. 28%). Within each class, conserved genes are less variable (Appendix Fig S3C); hence, sequence conservation provides additional information about variation constraints across genes.

Overall, these results suggest that expression variation is an orthogonal component to regulatory plasticity. Regulatory plasticity was previously defined based on promoter shape information alone, with broad promoter genes generally being more constitutive and narrow promoter genes being more condition-specific (Rach *et al*,

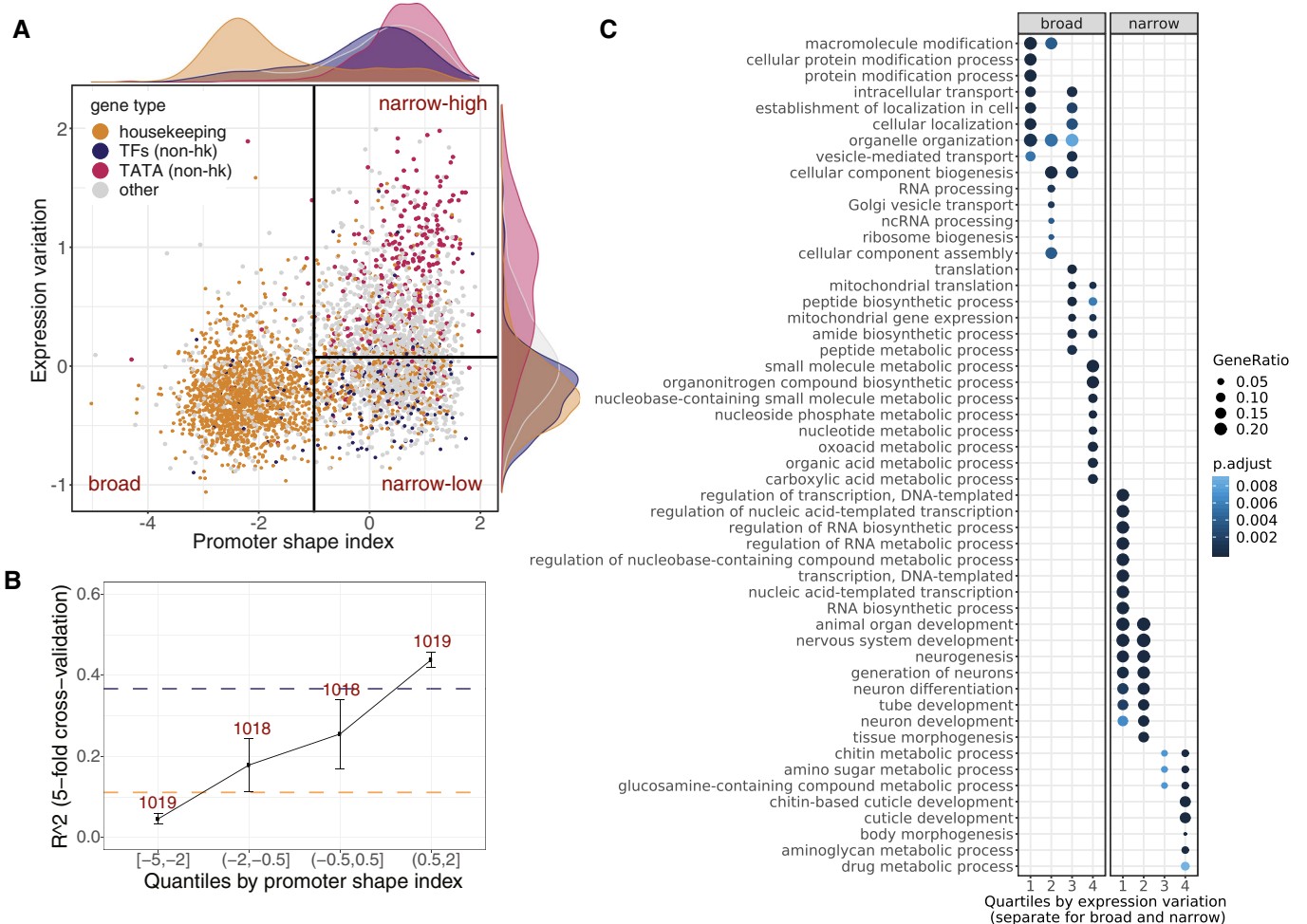

**Figure 3. Expression variation in broad versus narrow promoter genes reflects trade-offs between expression robustness and regulatory plasticity.**

A Genes separate into three groups based on their promoter shape index (*x*-axis) and expression variation (*y*-axis). Each dot represents a gene; colours indicate gene annotations: housekeeping (orange), non-housekeeping TFs (blue), non-housekeeping with a TATA-box (red) and other (grey). Distributions of promoter shape index and expression variation across gene groups are shown as density plots. Broad and narrow promoter genes are separated based on shape index threshold of −1 (vertical black line) as in Schor *et al* (2017). Narrow-low and narrow-high groups are separated based on the median expression variation of narrow promoter genes (horizontal black line).

B Performance to predict expression variation for genes split by quartiles of promoter shape index. Data are presented as mean ± SD ($R^2$ from fivefold cross-validation). Horizontal lines show performance (mean $R^2$ from fivefold cross-validation) on broad (orange) and narrow (blue) promoter genes separately. Numbers of genes in each quartile are indicated.

C GO term enrichment (biological process) of genes stratified by promoter shape and expression variation. Top GO terms ranked by *P*-value are shown (full list in Dataset EV6). *P*-values (Benjamini–Hochberg-corrected) and gene ratio from compareCluster function (R clusterProfiler package) are reported. Quartiles of expression variation (1—lowest, 4—highest) were calculated for broad and narrow promoter genes separately. Quantile intervals for broad and narrow promoter genes provided in Materials and Methods.

2009; Lenhard *et al*, 2012). Regulatory plasticity (constitutive vs. condition-specific genes) likely reflects sequence properties within the promoter region, while expression variation may reflect evolutionary constraints on expression robustness with essential and highly conserved genes being less variable. These findings indicate a partial uncoupling between expression variation across multicellular individuals in a controlled environment and variation across tissues/development, analogous to the uncoupling between plasticity and noise observed in yeast (Lehner, 2010), and suggest different mechanisms to control expression robustness for genes with ubiquitous versus condition-specific expression.

## Two classes of genes with low variation have distinct regulatory mechanisms

The results above indicate that the partial uncoupling of expression variation and expression plasticity could be achieved by distinct mechanisms to ensure expression robustness between different promoter architectures (broad/narrow). To explore this, we examined the most predictive features in relation to the different promoter types. Among the most significant promoter features is "#conditions with DHS (TSS-prox)" (Fig 2A), which is derived from a comprehensive tissue- and embryonic stage-specific atlas of open

chromatin regions (DHS data for 19 conditions) during a time-course of *Drosophila* embryogenesis (J.P. Reddington, D.A. Garfield, O.M. Sigalova, K. Aslihan, C. Girardot, R. Marco-Ferreres, R.R. Viales, J.F. Degner, U. Ohler & E.E.M. Furlong, unpublished data). The median number of developmental conditions in which a gene had a DHS site at its promoter was 18, 8 and 1 for broad, narrow-low and narrow-high genes, respectively (Fig 4A), thus highlighting again that the narrow-low and broad classes differ in their developmental plasticity. A similar trend was observed for related features, such as using a compendium of TF occupancy data during embryogenesis (Fig 4B), TSS-proximal TF peaks with motifs, or motifs alone (Appendix Fig S4A and B). To understand how these promoter-type-specific DHS patterns are regulated, we examined the 24 TFs that were predictive of expression variation in the full model (Dataset EV4, "*med_imp_var*" column). Broad promoter genes were generally strongly enriched for occupancy by ubiquitously expressed TFs, insulator proteins and chromatin remodelers (e.g. BEAF-32, MESR4, E(bx); Fig 4C, Dataset EV5; Fisher's exact test). The narrow promoters were enriched for occupancy by the Polycomb-associated developmental factors Trl and Jarid2, with stronger enrichment for the narrow-low group (Fig 4C). Among the narrow promoters, the narrow-low were enriched for Jarid2 and Trl with respect to narrow-high (Fisher's test odds ratio = 1.86 and 1.43, *P*-value < 1e-3). Interestingly, some of the TFs enriched in broad versus narrow promoters are still predictive of expression variation in the narrow-promoter-only model (e.g. MESR4, E(bx) and YL-1; Appendix Fig S4C), while the presence of "narrow" TFs, despite being associated with low variation in narrow promoters, had the opposite effect in the broad class (Fig 4C bottom right).

The next most predictive feature in our model is "Pol II pausing index" (Fig 2A), defined as the density of polymerases in the promoter region divided by the gene body length (Saunders *et al*, 2013; Fig 2A). Narrow-low genes have the highest pausing index (median of 40) followed by broad and narrow-high genes (10 and 7, respectively; Appendix Fig S4D). Consequently, Pol II pausing is strongly negatively correlated with expression variation in narrow promoters (Spearman's correlation ρ = −0.28, *P*-value < 1e-16), yet showed no significant relationship in broad (Fig 4D), again highlighting different mechanisms to confer robust expression. This may be partially explained by Trl, which can modulate the level of Pol II pausing (Tsai *et al*, 2016).

Among the most significant non-promoter features, our model identified distal regulatory complexity ("#TF motifs (dist)" and "#DHS peaks (dist)" in Fig 2A) and post-transcriptional events ("#miRNA motifs" and "#RBP motifs" in Fig 2A) as predictive of expression variation. For distal regulatory complexity, narrow-low had the highest number of associated distal regulatory elements (median of 6), defined as DHS within 10 kb of the TSS, followed by broad (4) and narrow-high (4) genes (Appendix Fig S4G). Consequently, the number of distal DHS is negatively correlated with expression variation in genes with narrow promoters (ρ = −0.22, *P*-value < 1e-16) while being uncoupled from variation for broad (Fig 4F). Similarly, narrow-low genes have a higher number of miRNA motifs in their 3′UTRs (median of 35) compared to broad (20) and narrow-high (14) genes (Appendix Fig S4E), which again was negatively correlated with variation in narrow promoter genes only (ρ = −0.31, *P*-value < 1e-16; Fig 4E). Similar results were obtained for the number of RNA-binding protein (RBP) motifs,

which have an effect for narrow, but not for broad, genes (Appendix Fig S4F).

In summary, these findings provide strong evidence that robustness in gene expression across individuals is conveyed by different mechanisms depending on the gene's promoter type: in broad promoter genes, robust expression is likely a result of a plethora of broadly expressed TFs that bind to the core promoter and keep the chromatin constitutively accessible, compatible with their housekeeping roles. Narrow promoter genes, in contrast, seem to be regulated by a smaller number of (narrow-specific) TFs, and their robustness is conveyed through mechanisms that involve Pol II pausing, distal regulatory elements and post-transcriptional regulation. This suggests that broad and narrow promoter types have distinct mechanisms to regulate expression variation that are not necessarily transferable. This is possibly related to the relatively higher regulatory plasticity required of the narrow-low genes. Our results generalize findings on 14 developmental control genes, showing that Pol II pausing at promoters is linked to more synchronous gene activation, thereby reducing cell-to-cell variability in the activation of gene expression (Boettiger & Levine, 2009). Similarly, miRNAs have been proposed to buffer expression noise (Schmiedel *et al*, 2015; preprint: Schmiedel *et al*, 2017). Our data put these findings in a more global context as part of a distinct mechanism for a particular promoter type.

We summarized these mechanisms as two indices based on the ranked averages of the corresponding features: *broad regulatory index* (number of TF peaks, motifs and conditions with DHS at the TSS) and *narrow regulatory index* (Pol II pausing index, number of distal DHS and miRNA motifs), respectively (Fig 4G), which nicely separate the three gene groups. Interestingly, we found no evidence for a specific noise-amplifying factor, except for the TATA-box. Yet, even for TATA-box genes, since they are depleted of all the aforementioned robustness features (Appendix Fig S4H), the observed high variation may result from a lack of robustness-conveying factors rather than the presence of a TATA-box.

## Expression variation can predict signatures of differential expression

In the analysis above, we identified two distinct mechanisms that regulate expression variation, which are directly encoded in the genome. In the following, we assess the impact of these findings for interpreting gene expression studies in general.

We postulate that the expression variation of a gene across individuals can be interpreted as its ability to be modulated by any random perturbation. If this holds true, we expect expression variation to be predictive of a gene's response to changes in the environment. To test this, we used an independent gene expression dataset from adult flies that were subjected to different stress conditions related to temperature, starvation, radiation and fungus infection (Moskalev *et al*, 2015). In agreement with our hypothesis, genes that are differentially expressed in at least one stress condition (DE) also have high expression variation in our embryonic dataset (Fig 5A, Wilcoxon test, *P*-value < 1e-16), as seen for both up- and downregulated genes (Appendix Fig S5A). Remarkably, this held true for every individual stress condition (Appendix Fig S5B).

Differentially expressed genes are moderately enriched for narrow promoters (63 vs. 52 % for non-DE genes, Fisher's test odds

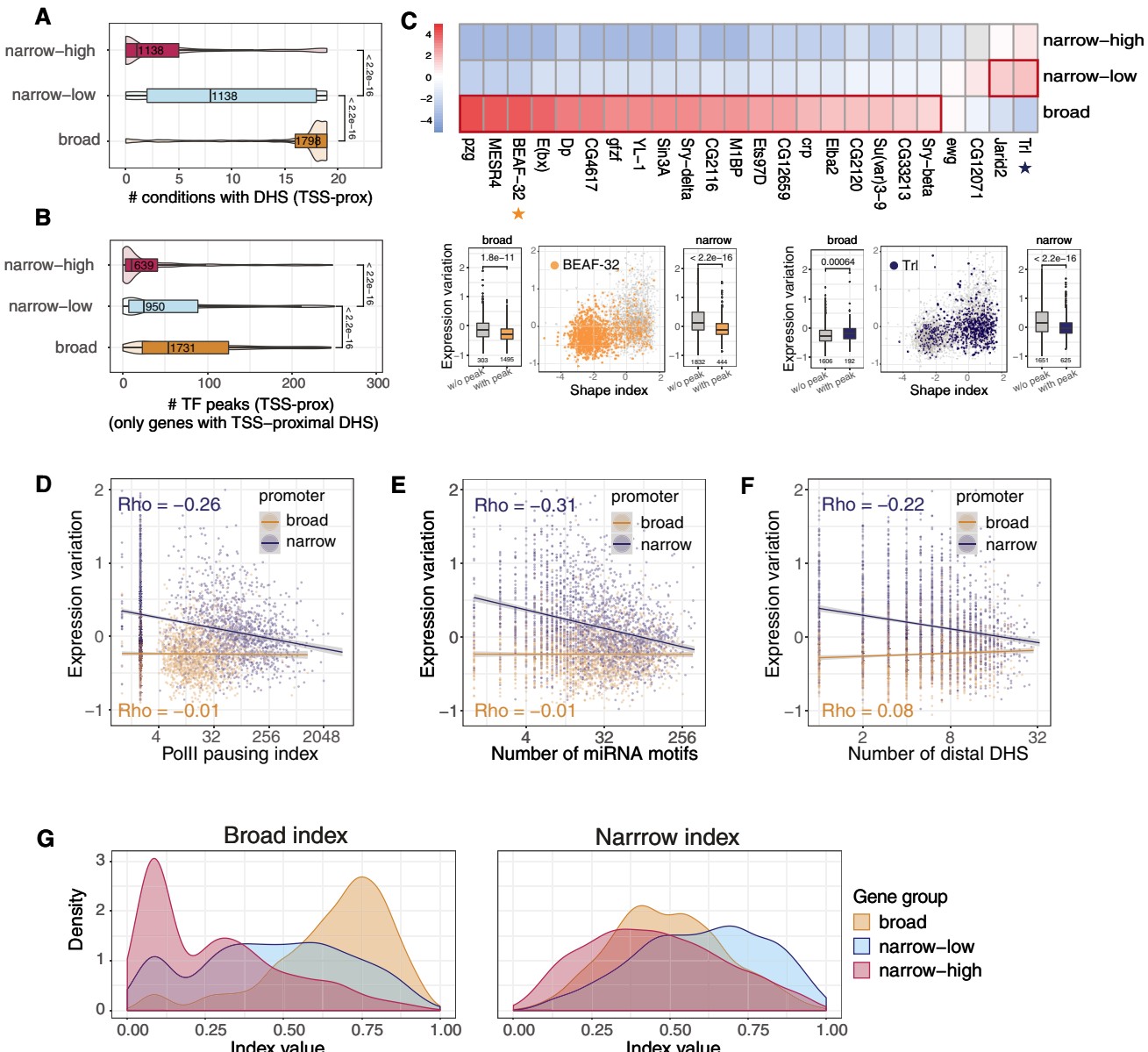

**Figure 4. Different regulatory mechanisms lead to expression robustness in genes with broad and narrow promoters.**

A, B   Chromatin accessibility—number of developmental conditions with a DHS at the promoter (A), Cohen's *d* = 1.0 for broad versus narrow-low (*d* = 0.8 for narrow-low vs. narrow-high), or number of different TF peaks (B), Cohen's *d* = 0.3 for broad versus narrow-low (*d* = 0.3 for narrow-low vs. narrow-high) overlapping TSS-proximal DHS for genes stratified into broad, narrow-low and narrow-high (defined in Fig 3A). *P*-values from Wilcoxon test. In (B), only genes with TSS-proximal DHS peak are considered. Boxplots: median as central band, the first and the third quartiles as lower and upper hinges, and the upper and lower whiskers extend from the hinge to the largest/smallest value at most 1.5 inter-quartile range, respectively. Numbers of genes in each group indicated.

C   *Top*: enrichment (odds ratio from Fisher's test) of ChIP peaks for 24 TFs in TSS-proximal DHSs of broad, narrow-low and narrow-high genes. Only TFs with predictive importance for expression variation (based on Boruta) were included. For each TF, Fisher's test was performed separately for each category versus all other. Colour = log2 odds ratio from Fisher's exact test (two-sided), grey = non-significant comparisons (adjusted *P*-value cut-off of 0.01, Benjamini–Hochberg correction on all 24x3 comparisons). Strong enrichments (odds ratio above 2) are outlined in red. *Lower panels*: the presence of BEAF-32 (left) and Trl (right) ChIP-seq peaks in TSS-proximal DHS, plotted in coordinates of promoter shape index and expression variation (same as Fig 3A). Each dot represents a gene (grey if TF peak is absent, blue for Trl, orange for BEAF-32). Boxplots on the left and right sides of the scatter plots compare expression variation (*y*-axis) of genes with and without ChIP-seq peak (*x*-axis) for broad and narrow promoter genes, respectively. Numbers of genes in each category are indicated. Boxplots: median as central band, the first and the third quartiles as lower and upper hinges, and the upper and lower whiskers extend from the hinge to the largest/smallest value at most 1.5 inter-quartile range, respectively. *P*-values from Wilcoxon test.

D–F   Relationship between polymerase pausing index (D), number of miRNA motifs in 3'UTR of a gene (E), number of TSS-distal DHS peaks (F) and expression variation for broad (orange) and narrow (blue) promoter genes. Each dot represents a gene, lines linear regression fits, ρ = Spearman's correlation coefficient.

G   Gene scores by two indices constructed as the normalized rank average of: number of embryonic conditions with DHS, number of TF peaks, number of TF motifs (broad regulatory index; left), and number of TSS-distal DHS, number of miRNA motifs, Pol II pausing index (narrow regulatory index; right). Colours correspond to broad (orange), narrow-low (blue) and narrow-high (red)) gene groups. *P*-values < 1e-09 for all pairwise comparisons of the distributions (Wilcoxon test).

ratio = 1.5, *P*-value = 1.5e-10). Within both narrow and broad promoter groups, DE genes are characterized by condition-specific chromatin accessibility at their promoter (Appendix Fig S5C), lower number of TSS-distal DHSs (Appendix Fig S5D), lower Pol II pausing index (Appendix Fig S5E) and a lower number of miRNA motifs (Appendix Fig S5F). Overall, DE genes showed lower regulatory complexity as reflected in our broad and narrow indices (Fig 5B and C).

To assess this association more systematically, we next examined whether the model for predicting expression variation could also identify differentially expressed genes. We trained a random forest model using our embryonic data to classify the top 30% versus bottom 30% of variable genes and used it to predict differential expression in adult flies subjected to different stresses (Materials and Methods). Remarkably, the model predicted differential

expression on the non-overlapping test set with an AUC of 0.65 and 0.74 when trained to predict embryonic variation for all genes and for narrow promoter genes, respectively (Fig 5D). This demonstrates that differential expression can be predicted based on a model trained for predicting expression variation. Since the model's performance was better when trained only on variation in narrow promoters, it is likely that the narrow-specific regulatory mechanisms, such as microRNA and enhancers, determine a gene's responsiveness to stress. This is also reflected by the strong differences in narrow index between DE and non-DE genes (Fig 5C).

To assess whether similar associations with expression variation exist for genes that are differentially expressed upon genetic perturbations, we collected data from 53 differential expression studies with genetic perturbations in *Drosophila* (Materials and Methods and Dataset EV8). For each study, we compared the mean

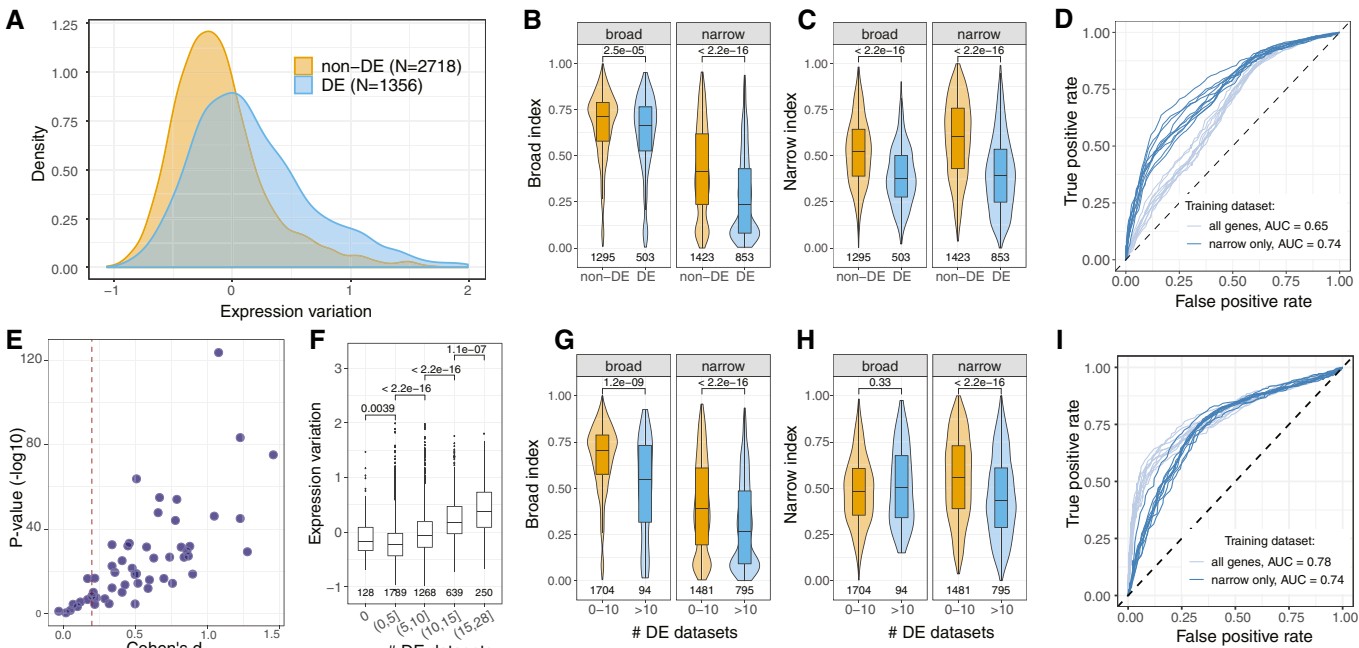

**Figure 5.  Expression variation can predict signatures of differential expression upon multiple conditions.**

A    Expression variation of genes differentially expressed (DE) upon any stress conditions in Moskalev *et al* (2015) compared to non-differentially expressed genes (non-DE), Cohen's *d* = 0.62.

B, C    Differences in scores by the broad and narrow indices (from Fig 4G) between DE and non-DE genes (from Fig 6A) split by promoter shape: broad index (B), narrow index (C), *P*-values from Wilcoxon rank test, numbers indicate number of genes. Cohen's *d*: (B) 0.25 and 0.60, (C) 0.75 and 0.90 for broad and narrow promoter genes, respectively. Boxplots: median as central band, the first and the third quartiles as lower and upper hinges, and the upper and lower whiskers extend from the hinge to the largest/smallest value at most 1.5 inter-quartile range, respectively.

D    ROC curves for predicting DE (from Fig 6A) with random forest models trained on expression variation (top 30% of variable genes vs. bottom 30% of variable genes) in all genes (light blue) or narrow promoter genes (dark blue). Models were trained and tested on non-overlapping subsets of genes in 10 random sampling rounds (all plotted). Median AUC values from 10 sampling rounds.

E    Difference in expression variation for DE versus non-DE genes in 53 datasets with different genetic perturbations (Dataset EV8). Cohen's *d* and *P*-value from Wilcoxon rank test for comparing DE versus non-DE genes are shown as scatter plot; each dot represents one dataset. Vertical dashed line indicates cut-off on minimal effect size (0.2).

F    Expression variation (*y*-axis) of genes found as DE in different numbers of genetic perturbation studies (groups on the *x*-axis), *P*-values from Wilcoxon rank test, numbers indicate number of genes. Boxplots: median as central band, the first and the third quartiles as lower and upper hinges, the upper and lower whiskers extend from the hinge to the largest/smallest value at most 1.5 inter-quartile range, respectively.

G–I    Same as (B–D) for comparison between genes that were found as DE in more than 10 studies with genetic perturbation versus genes DE in 0–10 studies. Cohen's *d*: (G) 0.86 and 0.41, (H) 0.14 and 0.43 for broad and narrow promoter genes, respectively (absolute values). Boxplots: median as central band, the first and the third quartiles as lower and upper hinges, and the upper and lower whiskers extend from the hinge to the largest/smallest value at most 1.5 inter-quartile range, respectively.

expression variation of all DE versus non-DE genes (union of conditions, if multiple conditions were tested) using Cohen's $d$ as the measure of effect size and Wilcoxon rank test for significance. For the majority of studies (83%, 44/53), DE genes were more variable compared to non-DE genes (Cohen's $d > 0.2$; Fig 5E). These findings, based on genetic perturbations, agree with the conclusions drawn from stress response experiments described above.

In addition, we observe a positive relationship between the number of studies in which a gene was differentially expressed and its degree of expression variation in our data (Fig 5F). One interpretation of this is that genes observed in only one specific perturbation may be more direct targets, and thus potentially more interesting to follow up than those that frequently change their expression regardless of the treatment. The latter may reflect a group of genes that is highly responsive to any environmental differences between test and control samples or stress induced by the genetic perturbation. Genes that were differentially expressed in multiple studies were strongly enriched in narrow promoters (89% of DE genes in more than 10 studies have narrow promoter versus 46% of DE genes in 0–10 studies, Fisher's test odds ratio = 9.7, $P$-value < 1e-16). Within the narrow promoter group, genes that were differentially expressed in more than 10 studies had significantly lower regulatory complexity, as indicated by various regulatory features (Fig 5G and H, Appendix Fig S5G–J). Finally, the model for predicting expression variation (top 30 vs. bottom 30% of variable genes) can identify genes differentially expressed in multiple experiments (DE in more than 10 datasets vs. DE 0–10 datasets) following the same methodology described above. The model was performed with AUC of 0.78 on all genes (mostly, classifying broad vs. narrow promoter genes) and AUC of 0.74 on narrow promoter genes only (Fig 5I).

Taken together, these findings indicate that expression variation across individuals is strongly linked to differential expression between conditions.

## Human promoter features predict both expression variation and differential expression

Given that gene expression variation across individuals can be predicted from genomic features in *Drosophila*, we next asked whether this holds true in humans, and whether the predictive features are conserved. We used high-quality RNA-seq datasets from the GTEx project comprising 43 tissues with data for at least 100 individuals (Lonsdale *et al*, 2013). For each tissue, we measured expression variation across individuals using the coefficient of variation corrected for mean–variance dependence, applying a similar approach as for *Drosophila* (Materials and Methods). Since gene expression variation values were highly correlated across all tissues (Appendix Fig S6), we also computed the mean of tissue-specific variations (mean variation) as potentially more robust metrics.

Since TSS-proximal features were the most predictive of expression variation in fly, we focused on promoter features to train the models (Materials and Methods). This included promoter shape, TF binding at the TSS, chromatin states and several sequence features (TATA-box, GC-content, CpG islands). To predict the mean expression variation, promoter shape and chromatin state features were aggregated across multiple tissues. In addition, we collated three tissue-specific datasets for muscle, lung and ovary by matching RNA-seq, CAGE and chromHMM datasets (Materials and Methods).

Based only on these features, random forest models were able to predict expression variation and level within each tissue to a similar extent as in *Drosophila* embryos (Fig 2F) with $R^2$ ranging between 0.38 and 0.46 for expression variation and between 0.19 and 0.24 for expression level (Fig 6A). Aggregating expression variation across tissues yielded even higher performance, with $R^2$ of 0.56 versus 0.31 for mean level across all expressing tissues. The overall performance was robust to changes in the numbers of samples including stratification by age or sex (Appendix Fig S8A).

The predictive features of expression variation in humans are highly overlapping with those in *Drosophila* (Fig 6B and C), and include promoter shape, TATA-box and the number of TFs binding to the promoter. An additional feature highly predictive of genes with low expression variation was the presence of CpG islands, in line with previous findings in single cells (Morgan & Marioni, 2018), while bivalent TSS state was predictive of high expression variation, in line with previous studies (Faure *et al*, 2017; Fig 6B and C). We also uncovered a number of transcription factors predictive of low variation, including GABPA, YY1 and E2F1 (84 predictive TFs in total, Dataset EV18). Similar to *Drosophila*, the presence of TSS-proximal peaks of all 84 predictive TFs was associated with reduced mean expression variation, again suggesting that high variation (in bulk RNA-seq) is due to a lack of buffering mechanisms rather than a specific mechanism for noise amplification. Extending the distance around the TSS did not improve the correlation between the presence of TF peaks and expression variation, indicating that the key regulatory information is already contained within the core promoter region (Appendix Fig S8B).

We next asked whether expression variation across individuals is predictive of differential expression in different conditions, similar to what we observed in *Drosophila*. For this, we used differential expression prior (DE prior), a metric that integrates more than 600 published differential expression datasets and reflects the probability of a gene to be DE irrespective of the biological condition tested (Crow *et al*, 2019). Indeed, DE prior is correlated with expression variation in all tissues (median Spearman's correlation ρ = 0.50) while being uncorrelated with expression level (Appendix Fig S6). A model trained to predict the top 30% versus bottom 30% of the most variable genes (based on the features predictive of mean expression variation) could predict DE prior with an AUC of 0.75 versus 0.85 when both training and testing are done on DE prior (Fig 6D, Materials and Methods), and predictive features for variation showed similar effects in DE prior (Fig 6E). This indicates that inherent promoter features can explain expression variation and the probability of differential expression to a similar extent—potentially due to partially overlapping underlying mechanisms.

Importantly, both expression variation and DE prior were significantly lower for essential genes while being higher for GWAS hits and common drug targets (Fig 6F, Appendix Fig S8C). Higher expression variation of the latter agrees with an interpretation that these genes are less buffered to withstand different sources of variation (Fig 1A) and hence are more likely to change in expression level upon different types of perturbations including genetic or environmental factors.

In summary, despite significant differences in promoter regions between humans and *Drosophila* (e.g. the presence of *Drosophila*-specific core promoter motifs, human-specific CpG islands, predominately unidirectional versus bidirectional transcription), promoter

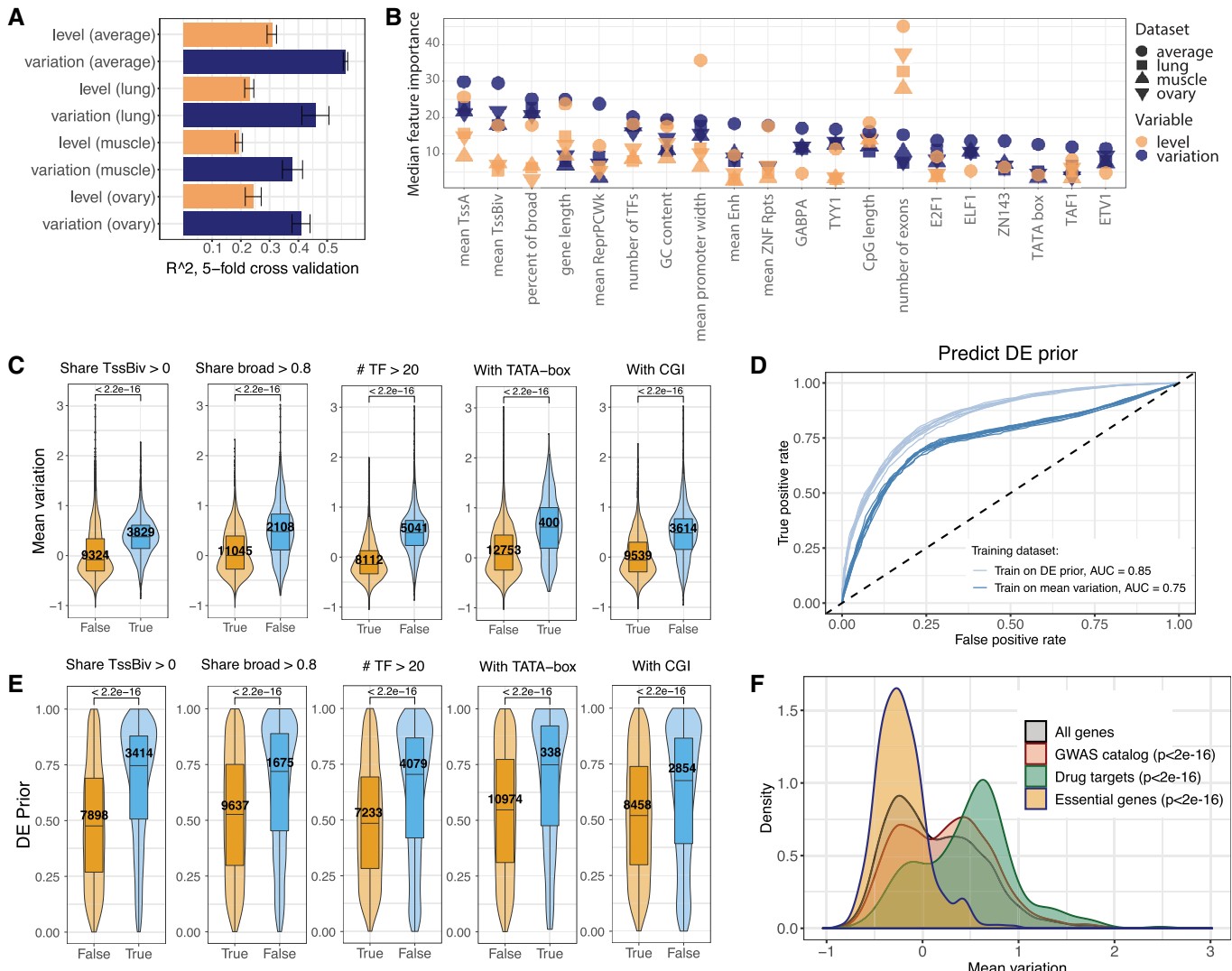

**Figure 6. Features in human promoters predict both expression variation and differential expression.**

A   Performance of random forest predictions ($R^2$) for expression level (orange) and variation (blue) trained on expression variation in tissue-specific RNA-seq (lung, ovary and muscle), as well as mean variation across 43 tissues (Materials and Methods). Data are presented as mean ± SD (fivefold cross-validation)

B   Top 20 features for predicting expression variation using Boruta feature selection. Features ordered by their importance for expression variation (blue), showing the corresponding importance for level (orange). Shapes indicate four different datasets (three tissues and mean variation).

C   Differences in expression variation for some of the top predictive features from (B). "Share TssBiv > 0" indicates genes that have "TSS bivalent" chromatin state (chromHMM, Materials and Methods) in at least one tissue. "Share broad > 0.8" indicates genes that have broad promoter in at least 80% of tissues where it is expressed. "#TF > 20" indicates genes with more than 20 different TF peaks in TSS-proximal region. "With TATA-box" and "With CGI" indicate the presence of TATA-box and CpG island in gene core promoter, respectively. *P*-values = Wilcoxon test, number of genes indicated. Cohen's *d*: 0.72 ("Share TssBiv > 0"), 0.93 ("Share broad > 0.8"), 1.43 ("#TF > 20"), 1.02 ("With TATA-box"), 1.03 ("With CGI"). Boxplots: median as central band, the first and the third quartiles as lower and upper hinges, and the upper and lower whiskers extend from the hinge to the largest/smallest value at most 1.5 inter-quartile range, respectively.

D   ROC curves for predicting DE prior (top 30% vs. bottom 30%) with random forest models trained on DE prior (light blue) and mean expression variation (dark blue). Models trained and tested on non-overlapping subsets of genes in 10 random sampling rounds (all plotted), with median AUC values indicated.

E   Differences in DE prior for some of the top predictive features from (B). "Share TssBiv > 0" indicates genes that have "TSS bivalent" chromatin state (chromHMM, Materials and Methods) in at least one tissue. "Share broad > 0.8" indicates genes that have broad promoter in at least 80% of tissues where it is expressed. "#TF > 20" indicates genes with more than 20 different TF peaks in TSS-proximal region. "With TATA-box" and "With CGI" indicate the presence of TATA-box and CpG island in gene core promoter, respectively. *P*-values = Wilcoxon test, number of genes indicated. Cohen's *d*: 0.71, 0.48, 0.5, 0.52, 0.35, respectively (same order as in (C)). Boxplots: median as central band, the first and the third quartiles as lower and upper hinges, and the upper and lower whiskers extend from the hinge to the largest/smallest value at most 1.5 inter-quartile range, respectively.

F   Mean expression variation of specific genes groups (GWAS hits, essential genes, drug targets) compared to the distribution of mean expression variation for all genes in the dataset. *P*-values = Wilcoxon test, Cohen's *d*: 0.2, 0.71 and 0.74 for GWAS catalogue, drug targets and essential genes, respectively (comparison to all genes).

features are highly predictive of expression variation in both species. Genes with high variation tend to also have differential expression across diverse conditions, and are significantly enriched in disease-associated loci (GWAS hits).

## Discussion

Our analysis suggests that expression variation across a population of multicellular genetically diverse individuals is gene-specific and can be explained by genetically encoded regulatory features, all highly correlated with core promoter architecture. Overall, we found that regulatory complexity positively correlates with robust gene expression. Yet, we identified two independent mechanisms that decrease expression variation depending on the core promoter architecture. Genes with broad core promoters in *Drosophila* were overall less variable and characterized by ubiquitously open chromatin and a high number of transcription factors (TFs) binding to the TSS-proximal region. In contrast, genes with a narrow core promoter had a much higher spread of expression variation, which was, in addition to TFs, modulated by regulatory complexity outside of core promoters (miRNAs, number of enhancers and Pol II pausing).

Remarkably, we found similar promoter-related features were predictive of expression variation across human individuals by applying the same predictive framework to tissue-specific RNA-seq datasets. This was surprising given the differences in promoter features between *Drosophila* and mammals, with higher heterogeneity within broad promoters and high regulatory importance of CpG islands (Lenhard *et al*, 2012; Haberle & Stark, 2018), and suggests that some core promoter properties are ancient features that reduce expression noise, which agrees with conclusion of previous studies (Carey *et al*, 2013; Metzger *et al*, 2015).

It is interesting to note that differences in expression variation within broad promoter genes are poorly explained by the extensive set of regulatory features examined here (Fig 3B). One explanation is that this is due to the overall low variability of broad promoter genes—broad promoters, for example, are more tolerant of the presence of natural sequence variation across individuals (Schor *et al*, 2017). However, we cannot exclude the possibility that expression variation of these genes is better explained by orthogonal regulatory mechanisms not considered in this study, such as mRNA degradation rates or post-transcriptional modifications. In addition, the weaker (though significant) correlation of enhancer complexity with expression variation might be the result of incorrect enhancer to target gene assignment, which is one of the current challenges in genomics. Beyond the scope of this study, a systematic analysis of post-transcriptional regulation and enhancer complexity on expression variation presents an interesting direction for future research, when such datasets become available.

Gene expression variation can arise from a multiplicity of stochastic, environmental and genetic factors, and defining the exact cause of expression variation in a particular experiment is likely an intractable task. Even for single-cell experiments, which can control for genetic and macro-environmental factors, there is ongoing debate as to whether the observed gene-specific expression variation can be explained by intrinsic (e.g. transcription bursting) or extrinsic (cell-to-cell variability) factors (Battich *et al*, 2015; Larsson *et al*, 2019; Foreman & Wollman, 2020), or whether these sources are indistinguishable (Eling *et al*, 2019). Yet, despite the differences in interpretation of the underlying sources of variation, there is a consensus that genes differ in their expression variation. Here, we found that gene expression variation, in bulk data from thousands of cells, was highly reproducible across different datasets, including developmental time-points in *Drosophila* and tissues in humans, and did not depend on the identity of the samples used. This suggests that gene expression variation, along with expression level, can be used as an informative readout of gene function and regulation in multiple biological contexts.

Interestingly, we recapitulated most of the regulatory features previously linked to expression noise in single-cell experiments (Boettiger & Levine, 2009; Perry *et al*, 2010; Ravarani *et al*, 2016; Faure *et al*, 2017; Morgan & Marioni, 2018; preprint: Schmiedel *et al*, 2017), despite the fact that the composition of variation sources is very different between bulk and single-cell experiments. A number of studies have proposed that robustness to stochastic noise and robustness to environmental and genetic variation are highly correlated (Ciliberti *et al*, 2007; Lehner, 2008; Kaneko, 2011). In line with this hypothesis, expression variation in bulk is predictive of single-cell noise in yeast (Dong *et al*, 2011) and gene expression variation across individuals in human tissue samples correlates with promoter strength and multiple epigenetic features (Alemu *et al*, 2014). Indeed, genes that have evolved mechanisms to buffer stochastic variation in the levels of their expression may also be insensitive to non-stochastic changes, including genetic and environmental variation, as the same mechanisms would constrain them both (Lehner, 2008).

In line with the above, it was recently shown that genes with DE across many experiments are generally predictable, and to a large extent seem to reflect some basic underlying biology of the genes rather than the specific conditions tested (Crow *et al*, 2019). Our results confirm and substantially extend this model—we show that the likelihood of a gene to be differentially expressed is highly correlated with the gene's expression variation (independent of expression level) and the corresponding predictive regulatory features, suggesting that the same mechanism confers robustness to different kinds of perturbations. Further, we observe that genes that are differentially expressed in multiple studies are highly variable across individuals. These results have important implications, as standard differential expression pipelines correct for variance dependence on the expression level (Love *et al*, 2014) but do not take any other gene-specific properties into account. Our findings suggest that considering gene-specific differences in expression variation may improve specificity and interpretability of differential expression results as it will distinguish between genes that have an inherent tendency to respond to any perturbation from those specific to the given experiment. Interestingly, we observed similar patterns for naturally occurring stress conditions and experimentally induced genetic manipulations. The similarity might result from the fact that genetic perturbation itself introduces stress to the organism and changes cellular environment, hence triggering non-specific response from more variable genes.

Finally, here we focused on the most general mechanisms robustly linked to gene expression variation regardless of the specific tissue identity or developmental stage. There is, however, accumulating evidence that changes in expression variation can be an important indicator of specific biological processes happening in an

organism. In particular, stochasticity of expression can differ by developmental stage, i.e. following an hourglass pattern in early development (preprint: Liu *et al*, 2019) or decreasing with cell fate commitment (Richard *et al*, 2016; Eling *et al*, 2018). On the other hand, an increase in expression stochasticity has been linked to ageing (Viñuela *et al*, 2018; Kedlian *et al*, 2019) and certain disease conditions (Zhang *et al*, 2015; Ran & Daye, 2017). Hence, combining information on expected gene expression variation with tissue or disease-specific data might provide additional insights into condition-specific gene regulation in complex biological systems.

## Materials and Methods

### Gene expression level and variation in *Drosophila* DGRP lines

#### Gene expression quantification

To quantify gene expression, we reprocessed the single-end strand-specific 3′-Tag-seq data (Cannavò *et al*, 2017) for 75 inbred wild *Drosophila* isolates from the *Drosophila melanogaster* Genetic Reference Panel (Mackay *et al*, 2012) at three time-points during embryonic development (2–4, 6–8 and 10–12 h after fertilization, 225 samples in total, each containing pool of approximately 100 embryos). Reads were trimmed using Trimmomatic v.0.33 software (Bolger *et al*, 2014) with the following parameters: -phred33 HEADCROP:7 CROP:43. Alignment to the dm6 genome version (dos Santos *et al*, 2015) was done with bwa v.0.7.17 aln (parameters: -n 5 -e 10 -q 20) and samse (parameters: -n 1) tools (Li & Durbin, 2010). Reads with mapping quality below 20 were removed using samtools view v1.9 (Li *et al*, 2009). Expression was quantified with HTSeq-count v.0.9.1 (Anders *et al*, 2015; parameters: -m intersection-nonempty -f bam -s yes -q -i Parent). PolyA sites were identified by reproducing the analysis of the polyadenylation dataset published in Cannavò *et al* (2017) after mapping the reads to the dm6 genome assembly. We observed a partial failure of strand specificity in generating the sequencing libraries: highly expressed polyA sites showed a corresponding anti-sense site. To remove these artefacts, we excluded polyA sites that were perfectly included in an antisense site. Reads that spanned both the last transcribed base and the subsequent polyadenylation tail allowed for single-base resolution identification of the cleavage site. We extended polyA sites 200 bp downstream or up to the nearest polyA site. To identify cleavage sites within our polyA sites, we produced strand-specific coverage tracks of the 3′-terminal base for each of the polyadenylation reads. Within each polyA region, we identified the major cleavage site as the genomic base with the highest 3′-terminal base coverage.

#### Expression data filtering and measuring expression variation

All samples selected for the analysis had high sequencing quality and were accurately staged, as described in original publication (Cannavò *et al*, 2017). Using principal component analysis on the expression counts from all 225 samples after applying variance stabilization transformation from DESeq2 (Anders & Huber, 2010), we confirmed that samples were clustered by developmental time-point (Appendix Fig S1A) and not sequencing batch (Appendix Fig S1B).

Expression counts from 225 samples were jointly normalized using size factor normalization from DESeq2 package (Anders & Huber, 2010). For each time-point, we calculated median expression

and coefficient of variation (CV, standard deviation divided by mean) for each gene across 75 samples. Genes with zero median expression were removed as non-expressed. The CV exhibited a strong negative relationship with median expression level (Fig 1B), which agrees with previous studies (Anders & Huber, 2010; Faure *et al*, 2017; Ran & Daye, 2017; Eling *et al*, 2018). To account for this relationship, we employed local polynomial regression (LOESS; loess function in R from stats library, degree = 1, span = 0.75; R Development Core Team, 2013) of the CV on the median expression, and used the residuals (resid_cv, residual coefficient of variation) in all subsequent analyses, referring to them as gene *expression variation*.

To check whether expression variation reflects expression heterogeneity (across samples) at any given expression level, we took the following approach: genes were grouped into 20 bins by their median expression level across 75 samples (separately for each time-point). Within each bin, genes were ordered by their residual CV (*x*-axis), and normalized expression counts for each sample were plotted on the *y*-axis (example of 10–12 h in Appendix Fig S1C). For almost all expression bins, the spread of expression values increased with higher residual CV, except bin 20 (top 5% genes by expression level) and to a lesser extent bin 1 (bottom 5%). Based on this analysis, the top and bottom 5% of expressed genes were excluded from the analysis.

We focused our analysis on the latest developmental stage (10–12 h) and removed genes that decreased in expression between 2–4 and 10–12 h after fertilization. This was done to reduce confounding effects of maternal mRNA degradation and focus on the stage when the zygotic genome is fully activated (both processes happening from 2 h post-fertilization onwards; Tadros and Lipshitz (2009)). In total, we excluded 3,651 genes, from which 90% were detected as maternally deposited (genes expressed in unfertilized eggs; see below). In addition, genes with the strongest decrease in expression (threefold or more) were highly enriched in cell cycle biological processes (Dataset EV7), and cell cycle is known to slow down at later developmental stages (Edgar & O'Farrel, 1989). Hence, we reasoned that variation of these genes might be strongly confounded by extrinsic factors (maternal mRNA degradation and cell cycle) that are not of particular interest for this analysis.

Overall, the following filtering steps were applied to the data, and the corresponding genes were excluded from the final dataset:

1   Genes with zero median expression level across samples (as non-expressed genes);
2   Genes falling into top and bottom 5% by expression level (as potential source of outliers);
3   Genes that decreased in expression between 10–12 and 2–4 h after fertilization (as maternal genes with role in early embryonic development and potential targets for maternal mRNA degradation)
4   Genes with missing values in the feature table (see below) unless the feature can be easily imputed, i.e. 0 for the absence of transcription factor motif

Hence, our final dataset included 4,074 genes at 10–12 h post-fertilization. The final measure of expression variation was calculated as described above on the final set of genes to avoid residual dependence on the expression level after filtering (Fig 1B, "*resid_cv*" column in Dataset EV2). Expression summary statistics

for all three time-points including expression variation calculated at several intermediate filtering steps are provided in Dataset EV3.

### Expression variation on the subsets of samples

To test robustness of expression variation to the selection of samples (and hence potential batch effects), we performed multiple rounds of sample subsetting. Our full dataset comprised 75 samples (75 DGRP lines). For a given subset size $N$, we randomly selected $N$ samples from the full dataset. Gene expression variation was calculated on this subset as described above (including fitting LOESS), and the Pearson correlation coefficient of the resulting variation values with the variation on the full dataset was recorded. Radom selection of samples was performed 100 times for each subset size ($N$ = 5, 10, 20, 30, 40, 50, 60, 70 and 74 samples). Mean and standard deviation of correlation coefficients upon 100 rounds of sampling for each subset size are shown in Fig 1C.

### Expression level and variation of neighbouring genes

For this analysis, we considered all pairs of genes located on the same chromosomes and with TSS-to-TSS distance below 100 kB. Gene pairs were binned into five quantiles based on the distance between their TSSs. Coordinates of the topologically associated domains (TADs) were taken from the high-resolution Hi-C in Kc cells (Ramírez et al, 2018). Genes were assigned to TADs based on their TSS coordinates, and for all pairs of genes, we defined whether they belong to the same TAD or span the TAD border. This resulted in 10 groups of gene pairs (five quantiles by distance in same/different TADs). In each group, we computed the Spearman correlation coefficient in expression levels and in expression variations between genes in pairs (correlation between two vectors: first gene in pair vs. second gene in pair, counting each pair only once). Results are shown in Appendix Fig S1E.

### Alternative measures of expression variation

As alternative measures of expression variation, we tested the following metrics:

1   sd_vst: standard deviation after applying variance-stabilizing transformation from DESeq2 package to remove mean–variance dependence (instead of taking LOESS residuals)
2   resid_sd: LOESS residuals from regressing standard deviation on median expression
3   resid_mad: LOESS residuals from regressing median absolute deviation on median expression
4   resid_iqr: LOESS residuals from regressing inter-quartile range (between 25th and 75th percentiles) on median expression

These measures were calculated on the final set of 4,074 genes at 10–12 h post-fertilization. Dependences on the median expression before and after correction for these measures are provided in Appendix Fig S2A. The Pearson correlation coefficients with expression variation measured by resid_cv are shown in Appendix Fig 2B.

### Compiling Feature table for *Drosophila* dataset

The full list of features used in this analysis is provided in Dataset EV1. The features were grouped into seven classes (column "*Feature class*" in Dataset EV1): Genetics, Gene type, Gene body, TSS, 3′UTR, Distal regulators and Gene context. Below are the more detailed descriptions of how individual features were generated. The final dataset is provided in Dataset EV2.

### Basic gene properties and functional annotations

We used Flybase v6.13 genome annotation to find gene length (length_nt), number of transcripts (n_transcripts) and number of exons (n_exons) for each gene. Number of exons was defined as total number of unique exons regardless of transcript isoforms. Next, we used several gene annotations from in-house or external sources to identify specific functional groups of genes. Ubiquitously expressed genes (is_ubiquitous) were defined based on BDGP database (Tomancak et al, 2002) as genes having ubiquitous expression pattern in at least one developmental stage (data available for *Drosophila* embryonic stages 4–6, 7–8, 9–10, 11–12 and 13–16). Maternally deposited genes (is_maternal) were defined as genes expressed in unfertilized eggs the vgn line of *Drosophila melanogaster* at 2–4 or 6–8 h after egg laying (Ghavi-Helm et al, 2019). Housekeeping genes (is_housekeeping) were defined following the methodology in Ulianov et al (2015) as genes expressed with RPKM > 1 in all samples from Graveley et al (2011). The list of transcription factors (is_tf) was taken from Hammonds et al (2013).

### Human orthologs for Drosophila genes

Human orthologs for *Drosophila melanogaster* genes were identified with DIOPT-DRSC Integrative Ortholog Prediction Tool (Hu et al, 2011), and two features provided by the tool were added for each gene—conservation score (conserv_score, continuous variable indicating confidence of ortholog prediction) and conservation rank (conserv_rank, discrete variable taking the following values: none, low, moderate and high). Genes with "high" conservation rank were referred to as "conserved with human" (e.g. Appendix Fig S3C).

### Genetics

Cis share (*cis*) was used as an estimate of the contribution of genetic variation to the total gene expression variation. To calculate it, we used LIMIX variance decomposition (preprint: Lippert et al, 2014) on the normalized expression matrix (three time-points combined) to assess the proportion of gene expression variation explained by cis genetic variation (defined here as sum of cis and (cis × environment) components from LIMIX), population structure (trans genetic variation) and time/environment. 3′UTR variant index (utr3_variant_index) was used to approximate 3′UTR sequence variation across DGRP lines and also to control for potential effect of mappability bias on gene expression variation (since expression was quantified from 3′-Tag-seq data). It was calculated using the following formula: (total number of variants in gene's 3′UTR × mean allele frequency of variants) / total width of 3′UTR peaks. The variant counts and variant allele frequencies were obtained from the DGRP freeze2 vcf file (Huang et al, 2014), considering only the 75 lines used in this study. The presence of eQTL (with_eQTL) indicates whether a gene has associated expression QTL identified in Cannavò et al (2017) on the same expression dataset, which is also used in this study.

### GC-content

GC-content was calculated using bedtools-2.27.1 nuc software (Quinlan & Hall, 2010) for nucleotide sequences of genes (gene_gc) and regions of −100/+50 bp around gene TSS annotations from Flybase v6.13 (tss_gc).

### Pausing index, promoter shape and promoter motifs

Polymerase II pausing index (defined as the density of polymerases in the promoter region divided by the gene body) in *Drosophila melanogaster* embryos was taken from Saunders *et al* (2013).

Promoter shape index was taken from Schor *et al* (2017). The index was calculated following the methodology from Hoskins *et al* (2011). In brief, promoter shape index is the Shannon entropy of the TSS distribution within a promoter:

$$\mathrm{SI} = 2 + \sum_{1}^{L} p_i \log_2 p_i,$$

where $p$ is the probability of observing a TSS at base position $i$ within the promoter, $L$ is the set of base positions that have at least one TSS tag, and TSS positions were identified using the aggregated CAGE signal for all time-points and 81 fly lines from the Drosophila Genetic Reference Panel (DGRP) at three developmental time-points (Schor *et al*, 2017). For each gene, we used the promoter shape index of the most expressed TSS cluster (major_shape_ind). Promoters of genes were classified as broad if the shape index of the most expressed TSS was below −1, and narrow otherwise (shape). The threshold is based on the bimodality of shape index distribution and was defined in the original publication (Hoskins *et al*, 2011). If any of alternative TSSs of a gene had shape different from the most expressed one, alt_shape feature took value of 1 (and 0 otherwise).

Position weight matrixes (PWMs) for 8 core promoter motifs (Ohler *et al*, 2002; Ohler, 2006) were scanned in −100/+50-bp region around annotated TSSs from Flybase v6.13 using fimo-4.11.3 software (Grant *et al*, 2011) with uniform background (–bgfile –uniform–), no reverse complement (–norc) and default *P*-value threshold (1e-4). Motifs were first scanned for the 5′-most TSS of each gene (start coordinate of genes in Flybase v6.13 annotation) and referred to as "ohler_maj.motif_name" (e.g. ohler_maj.TATA for TATA-box; 0/1 for motif absence/presence, respectively). In addition, motifs were scanned for TSSs of all transcripts for each gene (start coordinates of transcripts in gff annotation). If a motif was predicted for some of the transcript TSSs but not for the gene TSS, then the corresponding feature ohler_alt.motif_name was given a value of 1, otherwise 0.

### DNase hypersensitive sites

DNase hypersensitive sites (DHSs) in *Drosophila melanogaster* embryos were identified in J.P. Reddington, D.A. Garfield, O.M. Sigalova, K. Aslihan, C. Girardot, R. Marco-Ferreres, R.R. Viales, J.F. Degner, U. Ohler & E.E.M. Furlong (unpublished data). The experiment was conducted at five developmental time-points in whole embryo (2–4, 4–6, 6–8, 8–10 and 10–12 h after fertilization) and with tissue sorting (mesoderm/ myogenic mesoderm, neuroblasts/postmitotic neurons and other (double negative) at all time-points except 2–4 h; bin-positive (VM+) and bin-negative (VM−) mesoderm (marker for visceral muscles) at 6–8 h. This resulted in 19 experiments, which we refer to here as *DHS conditions*. Peaks called in all experiments were combined in a single table, and for each DHS, conditions when the site was accessible were recorded. Coordinates of DHSs from the combined table were lifted over from dm3 to dm6 genome version using UCSC liftOver-5.2013 tool (Kent *et al*, 2002). DHS standard deviation (also referred here as regulatory plasticity)

was defined in J.P. Reddington, D.A. Garfield, O.M. Sigalova, K. Aslihan, C. Girardot, R. Marco-Ferreres, R.R. Viales, J.F. Degner, U. Ohler & E.E.M. Furlong (unpublished data). Standard deviation was calculated on DHS signal across time and tissues (all DHS conditions except 2–4 h, VM+ and VM- DHS) after applying variance-stabilizing transform from DESeq2 (Love *et al*, 2014).

For each gene, we quantified a number of features related to DHSs in TSS-proximal (± 500 bp of TSS from gene annotation, feature class **TSS**) or TSS-distal (more than 500 bp and less than 10kB around annotated TSS, feature class **Distal regulators**) regions:

- Number of conditions with DHS (num_dhs_conditions.prox and num_dhs_conditions.dist) is the number of conditions (out of 19 in total) when there was a DHS detected in TSS-proximal or TSS-distal region, respectively.
- DHS tissue profile (dht_tissue_profile.prox and dhs_tissue_profile.dist) summarizes accessibility profile across tissues and takes the following values: (i) DHS present only in sorted tissues (any of mesoderm, neuroectoderm and double negative), (ii) present in whole embryo (WE), (iii) both in WE and in tissues.
- DHS time profile (dhs_time_profile.prox and dhs_time_profile.dist) reflects accessibility profile across developmental time-points: (i) DHS present only at early developmental time-points (2–4, 4–6 or 6–8 h after fertilization), (ii) present only at late developmental time-points (8–10 or 10–12 h after fertilization), (iii) present in at least one early and late time-point.
- The presence of ubiquitous DHS (dhs_ubiq.prox and dhs_ubiq_.dist) indicates the presence of ubiquitously accessible DHS in the corresponding genomic region. We consider DHS as ubiquitous if it was present in all three tissues at four developmental time-points where tissue sorting was done (12 conditions in total).
- Number of DHSs (num_dhs_any.prox and num_dhs_any.dist) is the total number of non-overlapping DHSs in the corresponding intervals present in any of the 19 conditions.

### DNA-binding proteins

Two hundred and eighty embryonic ChIP-seq datasets for various DNA-binding proteins (referred to as transcription factors (TFs) for simplicity though not all of them have reported transcription factor activity) were downloaded from modERN database (Kudron *et al*, 2018). Of note, for several TFs, ChIP-seq data are available either for several developmental time-points (Trl at 0–24, 8–16 and 16–24 h after fertilization) or for several experimental setups (chif-RA-GFP and chif-RB-GFP). When more than one dataset was available for a TF, they were included independently. For the analysis, we used peaks called according to the methodology from the original publication (IDR threshold of 0.01, optimal set). ChIP-seq peaks were overlapped with DHSs (single-base pair overlap required) using findOverlaps function from GenomicRanges package in R (Lawrence *et al*, 2013) resulting in 280 binary variables (1/0 for presence/absence of each TF) for each DHS. These data were then summarized for each gene's TSS-proximal and TSS-distal region resulting in the following variables:

- The presence of TF peak in TSS-proximal DHSs (280 variables with name format like modERN.tf_name.prox) and TSS-distal DHSs (280 variables with name format like modERN.tf_-name.dist); 1—peak present (any number of occurrences),

0—peak absent.

- Total number of different TF peaks (max. 280) overlapping TSS-proximal (num_tf_peaks.prox) and TSS-distal (num_tf_peaks.dist) DHSs.

### Transcription factor motifs

Three hundred and sixty-one PWMs for *Drosophila melanogaster* TF-binding motifs were downloaded from CISBP database (Weirauch *et al*, 2015). PWMs were scanned in the sequences of DHSs resized to 200 bp using fimo-4.11.3 (Grant *et al*, 2011) with uniform background (–bgfile –uniform–), with reverse complement (default), and default *P*-value threshold (1e-4). 343 PWMs with matches in our dataset were then assigned to 360 TFs using CISBP motif annotation (a few motifs were assigned to multiple genes) resulting in 360 binary variables corresponding to presence/absence of individual TF motifs. Similar to TF peaks, these data were then summarized for each gene's TSS-proximal and TSS-distal region resulting in the following variables:

- The presence of TF motif in TSS-proximal DHSs (360 variables with name format like cisbp.tf_name.prox) and TSS-distal DHSs (360 variables with name format like cisbp.tf_name.dist); 1—motif present (any number of occurrences), 0—motif absent.
- Total number of different TF motifs overlapping TSS-proximal (num_tf_motifs.prox) and TSS-distal (num_tf_motifs.dist) DHSs.

### Chromatin colours

Annotation of chromatin states (five states) was taken from Filion *et al* (2010). Coordinates of genomic regions assigned to different colours were lifted over from dm3 to dm6 genome version using UCSC liftOver-5.2013 tool (Kent *et al*, 2002) and overlapped with DHS coordinates. For each DHS, overlap with any of the colours by at least 1 bp was recorded. The results were aggregated by gene into five TSS-proximal (i.e. color_green.prox) and five TSS-distal (i.e. color_green.dist) binary features indicating presence/absence of the corresponding states.

### Annotated enhancers

We used several datasets of annotated enhancers from the following sources:

- Combined set of CAD4 enhancers (curated in-house list from various sources) and Vienna tiles (Kvon *et al*, 2014) lifted over to dm6 genome version;
- Combined set of cis-regulatory modules (CRMs) of mesoderm TFs (Zinzen *et al*, 2009) and cardiac TFs (Junion *et al*, 2012) lifted over to dm6 genome version.

Both datasets were first overlapped with DHS table, and the number of annotated enhancer elements in TSS-proximal and TSS-distal regions was added to the feature table following the same approach as above (features num_cad4_vienna_enh.prox, num_cad4_vienna_enh.dist, num_heart_meso_crm.prox and num_heart_meso_crm.dist).

### 3′ UTR features

Position weight matrixe of micro-RNAs (miRNAs) from miRBase (Kozomara & Griffiths-Jones, 2014; Kozomara *et al*, 2019) and RNA-binding proteins (RBPs) from CISBP-RNA (Ray *et al*, 2013) were downloaded from MEME v4 (Bailey *et al*, 2009) motif databases

(466 and 54 DNA-encoded PWMs from miRBase and CISBP-RNA, respectively). 3′UTRs were defined as the regions comprised between a major cleavage site (as defined above) and the closest annotated stop codon. PWMs were scanned in nucleotide sequences of the 3′UTRs using fimo-4.11.3 (Grant *et al*, 2011) with uniform background (–bgfile –uniform–), no reverse complement (–norc) and default *P*-value threshold (1e-4). Features for motif occurrences were named mirbase.motif_name and cisbp_rna.motif_name for miRNA and RBP motifs, respectively. The feature took a value of 1 for a gene if the corresponding motif was predicted for any of the annotated 3′UTRs of a gene, and 0 otherwise. The total number of unique miRNA and RBP motifs per gene was counted and included as num_mirna and num_rbp features, respectively.

Lists of genes that are putative targets of Pumilio (embryonic and adult data) and Smaug (embryonic data) RBPs were obtained from Gerber *et al* (2006) and Chen *et al* (2014), respectively.

For each gene, we calculated the mean UTR length at different time-points as the weighted mean UTR length between UTR isoforms (only 10–12 h 3′UTR lengths were used in the final feature table). We used the polyA site expression as weights in the mean calculation. Since length of 3′UTR was highly correlated with gene length (Spearman's correlation $\rho = 0.62$), utr3_length feature was calculated as actual 3′UTR length divided by gene length. Finally, 3′UTR length changes (log2 fold change) between different time-points (10–12 h vs. 6–8 h, 6–8 h vs. 2–4 h, and 10–12 h vs. 2–4 h) were calculated for each gene (utr3_l2fc_10vs6, utr3_l2fc_6vs2 and utr3_l2fc_10vs2 features).

### Genomic context features

The insulation score (ins_score_2_4 h and ins_score_6_8 h) was calculated based on Hi-C in-house data (unpublished) for *Drosophila melanogaster* embryos at 2–4 and 6–8 h after fertilization (in-nucleus ligation, whole embryo). To assign insulation scores to genes, we assigned the value of the nearest annotated TSS to each gene.

Coordinates of topologically associated domains (TADs) were taken from the high-resolution Hi-C in Kc cells from Ramírez *et al* (2018) and Hi-C in 2- to 4-h embryos (in-house data, unpublished). Each gene was then assigned to TAD from the two aforementioned annotations based on its TSS coordinate, and distance to TAD border and TAD size were recorded (dist_to_tad_border.ramirez, dist_to_tad_border.2_4 h, tad_size.ramirez and tad_size.2_4 h, respectively).

Gene density was calculated as number of genes in ± 1,000 bp and ± 20 kB from the TSS of each gene (num_genes.prox and num_genes_dist, respectively) based on Flybase v6.13 genome annotation.

### Broad and narrow indices

Broad and narrow indices were calculated based on the subset of features from the feature table. The broad index was composed of the following features (all TSS-proximal): number of conditions with DHS (num_dhs_conditions.prox), number of TF peaks (num_tf_peaks.prox) and number of TF motifs (num_tf_motifs.prox). The narrow index was composed of the following features: number of TSS-distal DHSs (num_dhs_any.dist), number of miRNA motifs (num_miRNA) and Pol II pausing index (PI). All features were first converted to ranks (random order for ties). Indices were calculated as simple averages of the corresponding features.

## Measuring expression level and variation in human tissues

### Genome version

We used Ensembl GRCH37/hg19 genome version downloaded from UCSC table browser (Kent et al, 2002; Haeussler et al, 2019) throughout the analysis. Sex chromosomes and non-standard chromosomes were removed from all subsequent analyses. For selecting the main transcript per gene, we used GRCH37/hg19 genome annotation downloaded from Ensembl website (Cunningham et al, 2019).

### Quantifying expression level and variation

Gene expression matrix (raw read counts) was downloaded from the GTEx Portal V7 (Lonsdale et al, 2013). Gene read count matrices per tissues were produced using GTEx sample details downloaded from GTEx website. Tissues with more than 100 samples (43 tissues in total) were chosen for further analysis (Dataset EV9). For each tissue, genes with 0 median counts were removed and expression counts were normalized using size factor normalization form DESeq2 package in R (Love et al, 2014). Median expression levels were calculated for each gene in each tissue and converted to log scale (natural logarithm) for subsequent analysis.

Next, we removed the top 5% of genes by median expression level as potential outliers, following the same reasoning as for Drosophila. Since distributions of gene expression in all tissues had long left tails, we set additional stringent threshold on lowly expressed genes (minimum median log-transformed normalized count of 5).

Gene expression variation was calculated on the final set of genes for each tissue following the same approach as for Drosophila; namely, gene expression variation was defined as the residuals from the local polynomial regression of coefficient of variation (CV) on the median expression (on the log scale, loess function in R from stats library, span = 0.25 and degree = 1). Gene expression levels and variations in all tissues are provided in Dataset EV10.

Mean expression variation for each gene was calculated as the mean of expression variations in all tissues where a gene was expressed (using final per-tissue tables that passed all filtering steps). Similarly, mean expression level was calculated by computing the mean of median expression levels in all tissues where a gene was expressed. Mean expression variation calculated in this way exhibited weak dependence on mean expression level (Spearman's correlation $\rho = -0.11$). To control for this effect, we also calculated "global mean variation" as the LOESS residuals of the mean CV on the mean expression level (calculated as above). This measure was highly correlated with mean variation (Appendix Fig S6) and showed similar results in the downstream analysis (results not shown).

### Feature tables for human dataset

Only TSS-proximal features and several gene properties (i.e. gene length and number of transcripts) were used to predict expression level and variation in human. The full list of features used in this analysis is provided in Dataset EV11. Most of the TSS-proximal features (TF peaks and chromatin states) were scanned in the $-500/+500$ bp of the main TSS of the genes (referred to as TSS-proximal regions), following the same approach as for Drosophila. Several features more strictly linked to the gene core promoters (promoter shape, TATA-box, CpG islands and promoter GC-content) were scanned in $-300/+200$ bp of the main TSS of the genes (referred to as core promoter regions).

### Gene properties

Number of transcripts, gene length, mean exon length, number of exons and exon length mean absolute deviation were calculated for each gene directly using hg19 genome annotation from Ensembl website (Cunningham et al, 2019). Transcript width was calculated for each transcript using the same file, and length of the main transcript was assigned to each gene.

### Promoter shape

CAGE data for 31 tissues (library size of about 10 M mapped reads or above, Dataset EV12) were downloaded from FANTOM5 project (Lizio et al, 2015) using CAGEr package in R (Haberle et al, 2015). On each dataset, we separately did a power-law normalization (Balwierz et al, 2009) using CAGEr package. TSSs with low count numbers (less than 5 counts) were removed. Next, we applied simple clustering method (distclu, maximum distance = 20) form CAGEr package on each dataset separately. Clusters with low normalized CAGE signal (sum of TSSs normalized signals of the cluster below 10–50 depending on the tissue) were removed. CAGE clusters were then assigned to genes by overlapping them with core promoter regions ($-300/+200$ bp around TSSs of all annotated transcripts). Clusters that did not overlap any core promoters were removed.

Next, we defined promoter shape for all CAGE clusters by using two commonly accepted metrics: promoter width and promoter shape index. Promoter width was calculated by using the interpercentile width of 0.05 and 0.95 following methodology from Haberle et al (2015). Promoter shape index (SI) was calculated by the formula as above (Drosophila section) proposed in Hoskins et al (2011).

For classifying promoters into broad and narrow types based on promoter width, we used the following approach: first, we did a linear transformation of promoter width values (actual value minus 1 divided by 10; for fitting a gamma distribution). On the transformed data, we fitted a gamma mixture model (2 gamma distribution), and parameters were trained using an EM algorithm (Dempster et al, 1977) implemented in the mixtools package in R (Benaglia et al, 2009). The threshold for classifying promoters as broad or narrow was selected by finding the point which best separates the two distributions. Following this approach, promoters with width above 10–15 bp were classified as broad, which was consistent across all tissues and agreed with earlier studies (FANTOM Consortium and the RIKEN PMI and CLST (DGT) et al, 2014). To classify promoters into broad/narrow using shape index, we fitted Gaussian mixture model (2 Gaussian distribution) to the data and selected the threshold separating the two distributions using the same approach as above. For the subsequent analysis, we used promoter width feature since it showed a more clear bimodal distribution in all tissues (example in Appendix Fig S7A–C) and is a more common metric in the analysis of mammalian promoters (Carninci et al, 2006; FANTOM Consortium and the RIKEN PMI and CLST (DGT) et al, 2014).

Each gene was then assigned the promoter width of its main transcript. If more than one CAGE cluster was present for a gene's main transcript, the cluster with the highest normalized CAGE signal was selected. Promoter width values for most of the genes were highly correlated across tissues (Appendix Fig S7D). Based on the tissue-specific shape data, we calculated two aggregated features for each gene. Mean promoter width (mean_promoter_width feature) was calculated as the mean of gene promoter widths in all tissues where it had CAGE signal (passing the filtering criteria defined above). The share of tissues where a gene had broad promoter (percent_of_broad feature) was calculated for each gene by dividing the number of tissues where the gene had a broad promoter by the total number of tissues where the gene had a CAGE signal.

### TATA-box motif

TATA-box motif coordinates were obtained from the PWMTools web server (Ambrosini et al, 2018): JASPAR CORE 2018 vertebrates motif library (Khan et al, 2018), P-value cut-off of 10-4, GRCh37/hg19 genome assembly. Motif coordinates were overlapped with gene core promoter regions (−300/+200 bp), and number of overlaps for each gene was recorded (TATA_box feature).

### Transcription factors

Transcription factor dataset (444 TFs, peaks with motifs, hg19 genome) was obtained from Vorontsov et al (2018). If several datasets were available for the same TF, the dataset with the best quality was selected. For each TF, the corresponding feature was calculated by overlapping the TF regions and gene TSS-proximal regions (−500/+500 bp) and counting the number of overlaps for each gene.

### Chromatin states

Chromatin state dataset (chromHMM core 15-state model with five marks and 127 epigenomes; Ernst and Kellis (2017)) was downloaded from the Epigenomics Roadmap project (https://egg2.wustl.edu/roadmap/web_portal/chr_state_learning.html). We considered 26 tissues (Dataset EV13). For each tissue, 15 features (one for each state, e.g. TssA or TssBiv) were obtained. Each feature was calculated by overlapping the corresponding state regions and gene TSS-proximal regions (−500/+500 bp) and counting the number of overlaps for each region. Finally, aggregated features (e.g. mean_TssA or mean_TssBiv) were calculated as the mean of feature values for each state over all 26 tissues.

### CpG Islands

CpG island (CGI) data for hg19 were downloaded from the UCSC Genome browser (Haeussler et al, 2019). For each CGI, these included CGI length (CpG_Length), number of CpG clusters (CpGNum) and number of GC dinucleotides (gcNum). The three corresponding features for each gene were calculated by overlapping CGI regions and gene core promoter regions (−300/+200 bp). When a gene's promoter did not overlap any CGI, the three features were assigned to 0. If multiple overlaps were present, CGI with the biggest overlap was considered for each gene.

### Promoter GC-Content

Promoter GC-content (GC_content) was calculated by using biostring package (Pagès et al, 2019) and BSgenome.

Hsapiens.UCSC.hg19 v1.4.0 in R in gene core promoter regions (−300/+200 bp).

### Compiling final feature tables

We collated three tissue-specific datasets for lung, muscle and ovary by combining the above promoter features and tissue-specific expression data (Datasets EV14–EV16). These tables included three types of features:

- Tissue-specific features (promoter width and chromatin states);
- Features aggregated across tissues (mean promoter width, percentage of broad and mean chromatin states; see above);
- Non-tissue-specific features (all other features, e.g. TATA-box or TF peaks)

These tables included genes that were expressed and had CAGE signal (passing the above filtering criteria in both datasets) in the corresponding tissues. For muscle tissue, the "Skeletal muscle male" dataset was used for tissue-specific chromatin states. The fourth feature table included only non-tissue-specific and aggregated features along with mean expression level and variation (Dataset EV17). This table is comprised of genes that are expressed and had CAGE signal in a least one of the analysed tissues. Expression variation was adjusted for the expression level on these final sets of genes in each table (see above).

### Essential genes, drug targets and GWAS catalogue

Essential genes (essential in multiple cultured cell lines based on CRISPR/Cas screens; Hart et al (2017)) and drug targets (FDA-approved drug targets (Wishart et al, 2018) and drug targets according to Nelson et al (2012)) were downloaded from the Macarthur Lab Repository (https://github.com/macarthur-lab/gene_lists). The GWAS dataset was downloaded from EBI GWAS catalogue (Buniello et al, 2019). Genes with GWAS associations within upstream regions or downstream regions were considered. These gene annotations were used in Fig 6F and Appendix Fig S8C, but not included in the prediction models. These gene annotations are provided in Dataset EV10.

## Predicting expression level and variation

### Random forest models for Drosophila embryos

Feature selection was done with the Boruta algorithm implemented in R (Kursa & Rudnicki, 2010) with the following parameters: P-value = 0.01, maxRuns = 500; Z-scores of mean decrease accuracy measure as importance attribute; Ranger implementation of random forest regression. Feature selection was done separately for several tasks: (i) predicting expression variation; (ii) predicting expression level (log-transformed values); (iii) predicting promoter shape index; and (iv–v) predicting expression variation and level in broad and narrow promoter genes separately. Median feature importance from 500 iterations was used as feature importance metrics. All features selected in at least one of the five settings listed above are provided in Dataset EV4 with the corresponding importance. Only selected features were used in random forest predictions and all downstream analysis.

For predicting expression level and variation, we ran random forest regressions using the mlr package in R (Bischl et al, 2016) with Ranger implementation of random forest (Wright & Ziegler, 2017);

default parameters: num.trees = 500, mtry = square root of the number variables. Model performance was reported with coefficient of determination ($R^2$) based on fivefold cross-validation (Fig 1D).

### Random forest models for human tissues

As above, we used random forest regression to predict expression level and variation in three tissues (lung, ovary and muscle; Datasets EV14–EV16), as well as mean expression level and variation (Dataset EV17). Feature selection and random forest regression were performed in the same way and with the same parameters as for *Drosophila* dataset. Boruta feature selection algorithm was used to select important features predictive of expression level (log-transformed) and expression variation in each of the four datasets. Feature importance scores are reported in Dataset EV18. Random forest regressions were run on the sets of selected features for the corresponding datasets. Model performance was reported with coefficient of determination ($R^2$) based on fivefold cross-validation (Fig 6A for performance in all four datasets).

### Testing robustness of the random forest models

#### Robustness tests for Drosophila dataset

We have run the following models to test robustness of our predictions to various potential confounding factors:

1.  Binning genes by their median expression level into five quantiles and rerunning variation prediction for each quantile separately (Fig 1E);
2.  Predicting alternative variation measures (see above): resid_sd, resid_mad, resid_iqr and sd_vst (Appendix Fig S2D);
3.  Binning genes by their median expression change between 10–12 and 6–8 h after fertilization into five quantiles and rerunning variation prediction for each quantile separately (Appendix Fig S2E);
4.  Binning genes by their promoter shape index into four quartiles and rerunning variation prediction for each quartile separately (Fig 3B).
5.  Training and predicting on different chromosomes (or chromosome arms), e.g. leaving out all genes on chr3L for testing the model trained on all other genes (Appendix Fig S2F).

For these tests, random forest regressions were run with the same parameters as above and on the set of features selected for variation prediction on the full set of genes. The performance of the models was measured with $R^2$ on the fivefold cross-validation in chromosomes (arms) 1–4 and on the holdout chromosome (arm) 5.

#### Robustness tests on human datasets

Since human gene expression datasets from GTEx project contain high sample heterogeneity (different ages, sexes, reasons of death, etc.), we have rerun prediction models on the following subsets of individuals (using sample metadata from GTEx website) for the lung tissue expression dataset:

- Only 20- to 39-year-old individuals;
- Only 40- to 59-year-old individuals;
- Only 60- to 79-year-old individuals;
- Only males;
- Only females
- Only violence group (as the reason of death);
- Only non-violence group (as the reason of death)

Gene expression variation and level were recalculated on the corresponding subsets of samples using the same methodology as above. Random forest regressions were rerun with the same parameters as above and on the set of features selected for the variation prediction on the full set of samples. Performance of the models was measured with $R^2$ on the fivefold cross-validation (Appendix Fig S8A).

### Collecting differential expression datasets in Drosophila

For the stress response experiments, we used lists of differentially expressed genes from Moskalev *et al* (2015). All experiments were conducted in adult *Drosophila melanogaster* flies (five-day-old males) and included the following stress condition: entomopathogenic fungus infection (10 CFU, 10 CFU), ionizing radiation (144 Gly, 360 Gly, 864 Gly), starvation (16 h) and cold shock (+4°C, 0°C, −4°C). In total, 1,356 out of our final set of 4,074 genes were detected as differentially expressed in at least one of the above stress conditions (DE) versus 2,718 non-DE genes.

For genetic perturbation experiments, we collected data from 53 differential expression studies with genetic perturbations in *Drosophila* (three datasets from the laboratory with mutated TFs involved in mesoderm formation and 50 datasets from Expression Atlas). Information about the collected datasets with PubMed IDs (PMID) of the corresponding publications was added as Dataset EV8. In total, there are 136 differential expression datasets for *Drosophila* deposited on Expression Atlas. For collecting data from Expression Atlas, we used the following criteria:

- Consider studies where the "Experimental factor" field includes keywords "genotype", "phenotype", "genetic modification" or "RNAi" to identify studies that include a genetic perturbation.
- Use absolute log2 fold change of 1 and FDR of 5% to define differentially expressed genes (default from Expression Atlas website)
- Only include datasets with at least 300 differentially expressed genes defined as specified above
- Exclude one dataset with more than 8,000 differentially expressed genes out of ∼12 thousand quantified (E-GEOD-28728) and two datasets where outlier samples are driving the signal (E-MEXP-1179 and E-GEOD-3069).

For each study, we took all differentially expressed (DE) genes (union of conditions, if multiple conditions were tested) and compared variation of DE versus non-DE genes in our set of genes using Cohen's *d* as the measure of effect size and *P*-value from Wilcoxon rank test (results provided in Dataset EV8). In total, 3,946 out of 4,074 genes in our final set were detected as differentially expressed in at least one experiment, and 889 genes were differentially expressed in more than 10 studies.

### Predicting differential expression in Drosophila

To test how well a model trained on expression variation can predict differential expression, we reformulated the variation

prediction into a classification task to predict the top 30% (class = 1) versus bottom 30% (class = −1) of genes ranked by their expression variation (our embryonic dataset) and used the trained model to predict DE (class = 1) versus non-DE genes (class = −1) from the stress response dataset. To avoid having the same genes in test and train sets, we undertook the following approach: Randomly sampled 50% of DE genes (678) and the same number of non-DE genes were set aside for the test set. From the remaining genes (after excluding test set—either all 2,718 genes or only genes with narrow promoters), the top 30% (class = 1) and bottom 30% (class = −1) of genes ranked by expression variation were used for training. The model was trained using random forest classification with default parameters (mlr package; Ranger implementation of random forest; default parameters: num.trees = 500, mtry = square root of the number variables). Training was performed on the features important for predicting expression variation on the full set of genes (see above, Dataset EV4) for expression variation. Testing was done on the same set of features for differential expression (1 for DE, −1 for non-DE). Performance on the test set was assessed by area under the ROC curve (AUC). Ten rounds of random sampling of genes were performed, and mean AUC was reported (Fig 5D).

For predicting differential expression in genetic perturbation experiments, we assigned genes detected as DE in more than 10 datasets to *class 1* and genes DE in 0–10 datasets to *class −1* and used the same approach as described above

## Predicting differential expression prior in humans

Differential expression prior data (DE Prior rank) were obtained from Crow *et al* (2019). Ensembl ids were converted to entrez ids by using BioMart package in R (Durinck *et al*, 2009).

We had information on both DE prior and mean variation (average of 43 tissue-specific variations across individuals; see above) for 11,312 human genes. As above, we reformulated the variation prediction into a classification task to predict the top 30% (class = 1) versus bottom 30% (class = −1) of genes ranked by their expression variation and used the trained model to predict top 30% (class = 1) versus bottom 30% (class = −1) of genes ranked by DE prior. Training and testing were performed on the set of features predictive of mean expression variation (Dataset EV18). Training and testing were done on the non-overlapping sets of genes using the following approach: first, we defined top prior (top 30% by DE prior) and bottom prior genes (bottom 30% by DE prior). Fifty percent of genes from both groups were randomly sampled and assigned to the test set. From the remaining genes, top 30% and bottom 30% of genes by mean expression variation were selected for the training set. The model was trained to classify top versus bottom variable genes (random forest classification with default parameters; mlr package in R, Ranger implementation of random forest). The trained model was then used on the test set to predict top versus bottom DE prior genes. Similarly, another model was trained and tested on classifying top versus bottom DE prior genes on the same training and test sets, respectively. Performance of the models on the test set was assessed by AUC. Ten rounds of random sampling of genes were performed, and mean AUC was reported (Fig 6D).

## Statistical data analysis and visualization

Data analysis in R was done using base, stats, MASS, rcompanion, psych, tidyverse, magrittr, data.table, ltm, yaml, Boruta, mlr, Ranger, GenomicRanges, DEseq2, CAGEr and rtracklayer packages. All plots were done in R using ggplot2, ggpubr, gridExtra, ggExtra, RColorBrewer and pheatmap libraries. Contour lines in Fig 2B–E represent 2D kernel density estimations (geom_density_2d with default parameters). *P*-values on the plots (Figs 2B–E, 4A–C, 5B and C, F–H, and 6C, E, F, Appendix Figs S3B and C, S4A–F, S5B–J) come from Wilcoxon rank test. Whiskers on the plots (Figs 1C–E, 2F, 3B, and 6A, Appendix Figs S2C and D, and S8A) indicate one standard deviation around the mean. In all boxplots (Figs 2B–E, 4A–C, 5B–C, F–H, and 6C and E, Appendix Figs S1F, S3B and C, S4A–F, and S5B–J), the central line corresponds to median, the lower and the upper hinges correspond to the first and the third quartiles, and the upper and lower whiskers extend from the hinge to the largest/smallest value at most 1.5 IQR (distance between the first and the third quartiles), respectively. All boxplots and violin plots were generated with geom_boxplot and geom_violin function from ggplot2.

### Effect sizes
Effect sizes for mean comparisons between groups of genes were estimated using Cohen's *d* (cohen.d in R from psych library, default parameters), defined as difference between two means divided by the pooled standard deviation. Empirical thresholds for Cohen's *d* of 0.2, 0.5 and 0.8 proposed in Cohen (1988) can be used to define small, medium and large effects, respectively. For boxplots and density plots comparing different gene groups, we report absolute values of Cohen's *d* in figure legends. In Fig 5E, actual values of Cohen's *d* are plotted on the *x*-axis (values above 0 indicate higher expression variation of differentially expressed genes compared to non-differentially expressed).

### Correlation analysis
Generally, we used the Spearman coefficient of correlation (R base) for comparing pairs of continuous variables or discrete variables taking more than two values (e.g. expression variation and promoter shape index or expression variation and conservation rank). In some cases, we used the Pearson correlation coefficient (R base) to compare variables that are on the same scale, e.g. expression variations at different time-points (Appendix Fig S1D) and expression variation upon subsetting of samples (Fig 1C). Finally, point-biserial correlation coefficient (R, ltm library) was computed between continuous and binary variables (e.g. expression variation and presence of TATA-box motif). Median expression levels were log-transformed before computing correlation.

### Gene Ontology enrichments
Gene Ontology (GO) enrichment tests were performed using clusterProfiler package in R (Yu *et al*, 2012). We used compareCluster function (*P*-value cut-off = 0.01) to find enriched biological processes (Fig 3C) and molecular functions (Appendix Fig S3A) in genes grouped by their promoter shape and expression variation. All genes from our final dataset (4,074) were used as background for GO enrichments. Genes with broad and narrow promoters were each split into four quantiles by their expression variation (x-axis labels in Fig 3C and Appendix Fig S3A indicate quantiles: from low

(1) to high (4) variation). Quantile intervals for broad promoter genes (1 to 4): $[-1.06, -0.444]$; $(-0.444, -0.266]$; $(-0.266, -0.0754]$; $(-0.0754, 1.89]$. Quantile intervals for narrow promoter genes (1 to 4): $[-0.98, -0.173]$; $(-0.173, 0.0751]$; $(0.0751, 0.416]$; $(0.416, 1.99]$. Full results of GO enrichment tests are provided in Dataset EV6.

### Fisher's tests

We used Fisher's exact test (R base package) to find enrichments of features in different gene groups in *Drosophila* dataset (broad, narrow-low and narrow-high). All tests were done for $2 \times 2$ contingency tables, and odds ratios and *P*-values provided by the test were recorded. We applied Benjamini–Hochberg correction to adjust *P*-values for the multiple testing and used an adjusted *P*-value threshold of 0.01 to define significantly enriched or depleted features.

For the analysis in Fig 3A, we tested enrichment of housekeeping genes, transcription factors and TATA-box promoter motifs in the following pairwise comparisons: (i) broad versus narrow, (ii) narrow-low versus two other groups, and (iii) harrow-high versus two other groups. *P*-values were corrected for the number of tests (9 comparisons).

For the analysis in Fig 4C, we tested enrichments of ChIP-seq peaks of 24 transcription factors in the TSS-proximal regions in the same comparisons as above. Twenty-five TSS-proximal TF features selected by the Boruta algorithm, including two ChIP-seqs for Trl (in embryos at 8–16 and 16–24 h after fertilization) from which the one with the overlapping time window was used (8–16 h). Since ChIP-seq peaks were first overlapped with DHSs before assigning to genes (see Features section above), we restricted the analysis of TF enrichments to the genes that have at least one DHS in their TSS-proximal regions. *P*-values were corrected for 72 comparisons (24*3). Log2-transformed odds ratios from these tests are shown in Fig 4C heatmap, actual values are provided in Dataset EV5.

Peaks of Trl and Jarid2 showed enrichments in both narrow-low and narrow-high genes (stronger enrichment in narrow-low; Fig 4C), which likely comes from strong depletion of these TFs at the TSSs of broad promoter genes. To control for that, we also tested enrichments of the same 24 TF peaks in three comparisons between gene groups: (i) broad versus narrow-low, (ii) broad versus narrow-high, and (iii) narrow-low versus narrow-high (also 72 comparisons for *P*-value correction).

Results from all Fisher's tests described above are provided in Dataset EV5.

## Data availability

Code used to generate the results is available at https://git.embl.de/sigalova/regulation_of_expression_variation. Expression variation, feature tables and result summaries are provided in Datasets EV1–EV18. Other datasets used to generate results can be provided upon request.

**Expanded View** for this article is available online.

## Acknowledgements

We thank all members of the Furlong and Zaugg laboratories for useful comments on the manuscript. The work was financially supported by the European Research Council (ERC advanced grant) agreement 322851 (CisRegVar) to EEMF. Open access funding enabled and organized by Projekt DEAL.

## Author contributions

EEMF, JBZ and OMS designed the study. OMS, AS and MF performed the computational analyses under supervision of JBZ and EEMF. OMS and AS made the figures. OMS, JBZ and EEMF wrote the manuscript with input from AS and MF. All authors read and approved the final manuscript.

## Conflict of interest

The authors declare that they have no conflict of interest.

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
