## [Review Process File · Molecular Systems Biology]

Predictive features of gene expression variation reveal mechanistic link with differential expression

Olga Sigalova, Amirreza Shaeiri, Mattia Forneris, Eileen Furlong, and Judith Zaugg

DOI: [10.15252/msb.20209539](https://doi.org/10.15252/msb.20209539)

Corresponding author(s): *Judith Zaugg (judith.zaugg@embl.de)*, *Eileen Furlong (furlong@embl.de)*

Review Timeline:

Submission Date:	25th Feb 20
Editorial Decision:	14th Apr 20
Revision Received:	27th May 20
Editorial Decision:	4th Jun 20
Revision Received:	26th Jun 20
Accepted:	30th Jun 20

Editor: Maria Polychronidou

Transaction Report:

14th Apr 2020

Manuscript Number: MSB-20-9539

Title: Predictive features of gene expression variation reveal mechanistic link with differential expression

Thank you again for submitting your work to Molecular Systems Biology. We have now heard back from the three referees who agreed to evaluate your study. Overall, the reviewers are quite supportive. They raise however some concerns and make suggestions for improving the study, which we would ask you to address in a revision.

As you will see below, the issues raised by the reviewers are rather clear and I think that there is no need to repeat the points listed below. Reviewer #3 suggests validating some of the findings experimentally e.g. by using CRISPR to modify the promoter of some candidate genes and evaluate whether expression variation is affected as predicted. We think that while these experiments would indeed increase the impact of the work, they are not mandatory for the acceptance of the study. That said, we would not be opposed to the inclusion of such data in case you have it already at hand.

Please let me know in case you would like to discuss any of the other issues raised.

REFEREE REPORTS

Reviewer #1:

The manuscript by Sigalova et al aims to identify the genomic features that predict gene expression levels across developmental stages, tissue types and in response to stress. This is a very interesting question and one that has not been addressed in a systematic way and using a comprehensive analytical framework before. The manuscript is of high value for the broad scientific community interested in understanding gene regulation. This study uses bulk RNA-sequencing from fly embryos and human tissues to identify regulatory elements, which can predict gene expression level and/or gene expression variation. From the fly embryos, they discover that ~100 genomic features are predictive of expression level and variation using a random forest model. For expression level, the most important predictor was whether the gene was a housekeeping gene. However, expression variation was most highly determined by promoter architecture. Furthermore, gene expression variation in fly embryos was predictive of differential gene expression in adult flies

exposed to a range of stresses. As a validation, the authors also considered gene expression in human tissues from the GTEx dataset, which showed broad similarities to what was observed in flies, despite significant differences in promoter architecture between the two species. Overall this is a solid manuscript with interesting and well-supported findings. My specific comments are below.

1. The analysis of the contribution of gene expression level to gene expression variation, although performed in different ways, misses to systematically consider the role of mean gene expression on gene expression variation. How does the model perform if mean gene expression is incorporated among the features tested?
2. Claims about natural selection are not fully supported by evolutionary analyses, thus they seem to be mostly speculative. Specific analyses to support the statements should be performed or these considerations should be moved to the discussion section.
3. Rather than annotating genes based on the GWAS catalog (mostly physical proximity), it would be more appropriate to annotate them based on TWAS results
4. What was used as background for cluster profiler? All expressed genes should be used, to account for biological processes representative of the tissue considered.

Reviewer #2:

In this manuscript, Sigalova et al. use machine learning approaches to identify genomic features that correlate with or predict variation in (bulk) gene expression between different isogenic *Drosophila* lines. After correcting for expression levels, which correlates strongly with expression variability, the authors characterize the features that correlate with gene expression variation. The predictive features are most strongly associated with promoter features and differ between housekeeping genes, TATA genes and genes with Pol II pausing. These features are also predictive of differential expression upon stress, as well as gene expression variability across human tissues, suggesting that expression variability is an intrinsic feature of promoters.

Many of the features that are found here to contribute to gene expression variation have been previously characterized in various species with various techniques. But while the individual identified features are not surprising, this study comprehensively characterizes features contributing to gene variation using an orthogonal approach to previous studies. Thus, the study provides a significant contribution to our understanding of gene expression variability.

Overall, the paper is well written and easy to follow. The experiments and the analysis support the major claims of the paper. I strongly recommend publication and have only minor points to be addressed.

Minor comments:

- 1) As shown in Figure 3b, the variation of broad promoter index genes are harder to predict than those from focused promoters. It is not clear to me why this is. Is this due to the normalization to gene expression levels at the beginning of the study or have the predictive features been missed for broad promoters? Would could they be?

2) The axes on the inserts in Figure 4c at the bottom are not clear. Please provide more labels and details in the figure legend.

Reviewer #3:

I read with great interest the manuscript by Sigalova et al. entitled "Predictive features of gene expression variation reveal a mechanistic link between expression variation and differential expression". In this manuscript the authors explore the genomic features that can explain how transcriptional precision is achieved among individuals within a population in both flies and humans. Using previously generated RNA-seq datasets, the authors found that promoter architecture plays a preponderant role in controlling gene expression variation. Moreover, they also uncovered that the same promoter features are important to determine whether genes are differentially expressed upon stress. This is an interesting study that extends and generalizes some previous observations based on much smaller datasets. Importantly, the authors' findings can help in the important and difficult task of uncovering disease-associated non-coding variants as well as Gene-Environmental interactions implicated in human disease. With this said, I have some comments/suggestions that I hope the authors can address and that I think could improve the manuscript:

- Are the same genomic features predictive of transcriptional precision among individuals also applicable to single cells within a tissue or cell population?. The authors could analyse some of the many single-cell RNA-seq datasets recently generated in whole embryos or individual tissues to address this question.
- The weaker effects of distal DHS over promoter architecture in the control of transcriptional precision could be partly attributed to the assignment of DHS to the incorrect target genes. To minimize these miss-assignments, the authors could take into account co-occurrence of DHS and target genes within the same TADs, compatibility rules between genes and enhancers as described by the Stark lab, similarity in chromatin states, etc.
- Throughout the manuscript there are many comparisons between the three major groups of genes with different promoter types. Many of these comparisons are presented as boxplots together with p-values that measure whether the differences between groups are statistically significant. I think that these plots should also include "Effect size" measurements in order to better illustrate the magnitude of the differences between groups. Given the large datasets being compared, significant p-values are probably "easy" to obtain, which does not necessarily mean that the observed differences are neither large in magnitude nor in biological relevance.
- The link between expression variation and differential expression is interesting but should be more thoroughly investigated, since only differential expression in response to environmental stress has been considered. It is known that transcriptional responses to different environmental insults share a significant fraction of genes. Many of these shared "stress response" genes are up-regulated regardless of the insult: is this also true for differentially expressed genes displaying high expression variation?. In other words, is expression variation preferentially associated with gene activation or also with silencing?. On the other hand, the authors should also consider other differential gene expression datasets in which changes in signalling conditions (e.g. treatment with morphogen agonists or antagonists) or genetic perturbations (e.g. KO for a transcription factor) are used

instead. Can the authors still observe a clear link between differential expression and expression variation?

- Finally, the manuscript would also improve if some of their major findings and claims are experimentally validated. For example, the authors could use CRISPR technology to modify the promoter of some candidate genes and evaluate whether expression variation is affected as predicted by the authors. Although this might not be essential for publication, it would increase the impact of the authors' claims.

We thank all three reviewers for their very encouraging and constructive suggestions, which have greatly improved the manuscript. We provide a Point-by-point response to each of the reviewers comments below, which we tried to address in full.

Overall changes to the manuscript

We have:

- included a comparison of the predictions of expression variation (R^2) with and without median expression level as a feature in the model (new Appendix Fig. S2c);
- changed the background set of genes used in the GO term enrichment analysis, replacing panels Fig. 3c and Appendix Fig. S3a;
- added Cohen's d (defined as difference between two means divided by a standard deviation) as a metric of effects sizes to the figure legends associated with the boxplots in Fig. 2b-e, 4a-b, 5a-c,g-h, 6c,e,f and Appendix Fig. S3c, S4a-f, S5c-f,g-j, S8c;
- performed an extensive analysis of the links between expression variation and differential expression using genetic perturbation data collected from 53 different differential expression studies with genetic perturbations in *Drosophila*;
- rephrased statements related to positive selection and evolution;
- added discussion points about predictions in broad promoter genes and enhancer to gene assignments;
- addressed specific questions of reviewers (see below), and rephrased text as suggested.

Reviewer #1:

The manuscript by Sigalova et al aims to identify the genomic features that predict gene expression levels across developmental stages, tissue types and in response to stress. This is a very interesting question and one that has not been addressed in a systematic way and using a comprehensive analytical framework before. The manuscript is of high value for the broad scientific community interested in understanding gene regulation. This study uses bulk RNA-sequencing from fly embryos and human tissues to identify regulatory elements, which can predict gene expression level and/or gene expression variation. From the fly embryos, they discover that ~100 genomic features are predictive of expression level and variation using a random forest model. For expression level, the most important predictor was whether the gene was a housekeeping gene. However, expression variation was most highly determined by promoter architecture. Furthermore, gene expression variation in fly embryos was predictive of differential gene expression in adult flies exposed to a range of stresses. As a validation, the authors also considered gene expression in human tissues from the GTEx dataset, which showed broad similarities to what was observed in flies, despite significant differences in promoter architecture between the two species. Overall this is a solid manuscript with interesting and well-supported findings. My specific comments are below.

Thank you for this very encouraging synopsis of our manuscript and for seeing the novelty of our work

1. *The analysis of the contribution of gene expression level to gene expression variation,*

although performed in different ways, misses to systematically consider the role of mean gene expression on gene expression variation. How does the model perform if mean gene expression is incorporated among the features tested?

Authors' response: We agree with the reviewer that this is an important point to discuss. We have now run the model to predict expression variation including median gene expression as feature and found that there is a slight improvement in the random forest performance ($\Delta R^2=0.07$). Furthermore, expression level was identified among the most important features based on feature importance score, despite the fact that expression level and variation are globally uncorrelated by construction (Pearson $R=0$, Spearman $\rho=0.04$, Fig 1b). This may be due to residual non-linear relationships between gene expression level and variation. In addition, many features predictive of expression variation are also predictive of expression level (albeit to varying degrees), indicating that both properties are implicitly linked and incorporates regulatory information from an overlapping set of features. We initially decided to exclude expression level from the features to predict expression variation to avoid confounding effects since expression level and variation are linked by construction (taking LOESS residuals) and rather report features predictive of each property separately (Fig.2). However, we agree that this point should be examined and discussed in the manuscript.

Authors' action: We have now included a comparison of R^2 for predictions of variation with and without median expression level as a feature (new Supplementary Fig. 2c, see below) and added the following text to the Results section on (p8, lines 152-157):

“Including gene expression as a feature lead to a slight improvement in the random forest performance ($\Delta R^2=0.07$, Appendix Figure S2c), even though expression level and variation are globally uncorrelated by construction (Fig 1b). To avoid any confounding effects, we decided to exclude expression level from the features to predict expression variation, and instead report results for predicting expression level and expression variation side by side.”

2. Claims about natural selection are not fully supported by evolutionary analyses, thus they seem to be mostly speculative. Specific analyses to support the statements should be performed or these considerations should be moved to the discussion section.

Authors' response: We agree with the reviewer that our paper lacks specific evolutionary analysis. We have rephrased the corresponding statements and now refer to earlier literature addressing this topic instead of deriving conclusions from our own data.

Authors' action: We have rephrased the following sentence on page 15 (lines 305-308):

*"Enrichment of low-variable genes in either housekeeping (broad) or developmental (narrow-low) functions may reflect selection pressure to reduce expression noise in genes essential for viability and development, **as suggested by previous studies (Lehner 2008; Fraser et al. 2004; Metzger et al. 2015).**"*

And removed the sentence referring to natural selection from the following paragraph on page 28 line 601 (strikethrough text):

"Importantly, both expression variation and DE prior were significantly lower for essential genes, while being higher for GWAS hits and common drug targets (Fig. 6f, Supplementary Fig. 8c). Higher expression variation of the latter agrees with an interpretation that these genes are less buffered to withstand different sources of variation (Fig 1a) and hence are more likely to change in expression level upon different types of perturbations including genetic or environmental factors. ~~Hence, expression variation across individuals likely captures differences in selection pressure and cost-benefit trade-offs between expression precision and plasticity.~~"

3. Rather than annotating genes based on the GWAS catalog (mostly physical proximity), it would be more appropriate to annotate them based on TWAS results

Authors' response: we agree with the reviewer that GWAS data does include a physical proximity bias. It would be interesting to compare our data with TWAS results, but unfortunately, we are not aware of an extensive catalogue of TWAS studies that could be used for this. Going forward, there will also be other lists of genes that will be interesting to compare to the variation of expression. To facilitate this, we will make our expression variation measurement available as a well-formatted supplementary table so that anyone can easily do these comparisons with additional lists of genes.

4. What was used as background for cluster profiler? All expressed genes should be used, to account for biological processes representative of the tissue considered.

Authors' response: We thank the reviewer for this comment. We have indeed used all genes from the org.Dm.eg.db database as background in our initial analysis, which we have changed now. However, instead of using all expressed genes (as the reviewer proposes) we now used our final set of expressed genes (i.e. after passing all filtering steps, described in Methods), which leaves 4074 expressed genes at 10-12h), as we felt this is a more appropriate background. Importantly, the change of background did not change the major conclusion from the analysis.

Authors' action: We have updated the results of the GO enrichment analysis in Datasets E V 6-7 and replaced panel *c* in Figure 3 (see below) and panel *a* in Appendix Figure S3.

In accordance with this, we have updated text on page 14 line 286:

*"Interestingly, when we group genes from the two promoter classes into quartiles based on their variation we find very specific functions enriched among them: the broad class is strongly enriched for housekeeping genes (Fishers' test odds ratio, OR=15.0, p-value<1e-16, Datas et EV5) and GO terms related to basic cellular processes (cellular transport, secretion, and DNA/RNA biogenesis), with the exception of the top 25% most variable genes within the group being **strongly enriched in housekeeping metabolic processes** (Fig. 3c, Appendix Figure S3a, Dataset EV6)."*

Reviewer #2:

In this manuscript, Sigalova et al. use machine learning approaches to identify genomic features that correlate with or predict variation in (bulk) gene expression between different isogenic Drosophila lines. After correcting for expression levels, which correlates strongly with expression variability, the authors characterize the features that correlate with gene

expression variation. The predictive features are most strongly associated with promoter features and differ between housekeeping genes, TATA genes and genes with Pol II pausing. These features are also predictive of differential expression upon stress, as well as gene expression variability across human tissues, suggesting that expression variability is an intrinsic feature of promoters.

Many of the features that are found here to contribute to gene expression variation have been previously characterized in various species with various techniques. But while the individual identified features are not surprising, this study comprehensively characterizes features contributing to gene variation using an orthogonal approach to previous studies. Thus, the study provides a significant contribution to our understanding of gene expression variability.

Overall, the paper is well written and easy to follow. The experiments and the analysis support the major claims of the paper. I strongly recommend publication and have only minor points to be addressed.

Thank you for seeing the value and contribution of our work, and for the constructive suggestions below

Minor comments:

1) As shown in Figure 3b, the variation of broad promoter index genes are harder to predict than those from focused promoters. It is not clear to me why this is. Is this due to the normalization to gene expression levels at the beginning of the study or have the predictive features been missed for broad promoters? Would could they be?

Authors' response: Well spotted – we agree, this is an interesting question. Indeed, the random forest performance is lower for broad promoter genes. We are not sure of the reason for this, but it could be several. First, broad promoter genes are on average less variable with a smaller range of variation values (Fig 3a). In the narrow promoter genes, we mostly explain the difference between high-variable vs. low-variable group, and not the variation within the low-variable group (i.e. our model performed less well to predict variation within the lower half of narrow promoter-genes). Second, even though we aimed to compile a comprehensive set of gene regulatory features, there are still some additional factors that potentially influence expression variation and for which there is actually very little genomic data – in particular, mRNA degradation rates and post-transcriptional modifications, for which there is unfortunately no comprehensive data available. Thus, since broad and narrow promoters are characteristic of different types of genes, potentially with very different regulatory programs, we can't exclude the possibility that their variation is explained by an orthogonal set of regulatory features that we could not include here due to lack of data.

Author's action: We have added the following paragraph about this in the Discussion section on pages 30-31 lines 654-661:

"It is interesting to note that differences in expression variation within broad promoter genes are poorly explained by the extensive set of regulatory features examined here (Fig 3b). One explanation is that this is due to the overall low variability of broad promoter genes - broad

promoters, for example, are more tolerant of the presence of natural sequence variation across individuals (Schor et al. 2017). However, we cannot exclude the possibility that expression variation of these genes is better explained by orthogonal regulatory mechanisms for which there is currently no data available, such as mRNA degradation rates or post-transcriptional modifications."

2) The axes on the inserts in Figure 4c at the bottom are not clear. Please provide more labels and details in the figure legend.

Author's action: We added labels to the boxplots in the insert plots and explanation in the Figure 3c legend:

"Lower panels: Presence of BEAF-32 (left) and Trl (right) ChIP-seq peaks in TSS-proximal DHS, plotted in coordinates of promoter shape index and expression variation (same as Fig. 3a). Each dot represents a gene (grey if TF peak is absent, blue for Trl, orange for BEAF-32). Boxplots on the left and right sides of the scatter plots compare expression variation (y-axis) of genes with and without ChIP-seq peak (x-axis) for broad and narrow promoter genes respectively. Numbers indicate number of genes in each category."

Reviewer #3:

I read with great interest the manuscript by Sigalova et al. entitled "Predictive features of gene expression variation reveal a mechanistic link between expression variation and differential expression". In this manuscript the authors explore the genomic features that can explain how

transcriptional precision is achieved among individuals within a population in both flies and humans. Using previously generated RNA-seq datasets, the authors found that promoter architecture plays a preponderant role in controlling gene expression variation. Moreover, they also uncovered that the same promoter features are important to determine whether genes are differentially expressed upon stress. This is an interesting study that extends and generalizes some previous observations based on much smaller datasets. Importantly, the authors' findings can help in the important and difficult task of uncovering disease-associated non-coding variants as well as Gene-Environmental interactions implicated in human disease.

We thank the reviewer for these constructive comments and for the exciting outlook from our study

With this said, I have some comments/suggestions that I hope the authors can address and that I think could improve the manuscript:

- Are the same genomic features predictive of transcriptional precision among individuals also applicable to single cells within a tissue or cell population?. The authors could analyse some of the many single-cell RNA-seq datasets recently generated in whole embryos or individual tissues to address this question.

Authors' response: This is an interesting question. We expect yes, as our study using bulk data has uncovered many properties that were identified in individual single cell studies, such as a TATA box, Pol II pausing etc. We agree with the reviewer that it would be interesting to address this directly. Please note that direct comparisons of bulk and single cell data are complicated by the differences in composition of expression variation (e.g. intrinsic and extrinsic noise). In particular, most of the existing single cell studies focus on identifying diversity of cell types and tissues in heterogeneous samples and therefore the main source of expression variability is tissue- and cell type-specific genes. In addition, correction for mean-variance dependence is more complicated in single-cell studies due to the low read counts per cell and dropout effects.

To partially address this question, we used a single-cell dataset in human embryonic stem cells (hESCs) from (Morgan and Marioni 2018) that was acquired to ask questions about expression noise (i.e. a clonal population of a single cell type), where variation has already been calculated. We used expression variation from this dataset (rCV^2 , referred to as expression noise) to compare it with mean expression variation from our study (mean of expression variations across individuals in different GTEx tissues) and to predict variation across single cells using our set of features. We found that the mean expression variation was moderately, yet significantly, correlated with expression noise (Spearman $\rho=0.27$). With our set of features, expression noise (log-transformed values) can be predicted with R^2 of 0.17 with a random forest model (5-fold cross-validation). Finally, we find that similar features are predictive of expression variation in bulk and in single cells, despite the fact that correlations of features with single cell noise are much lower than those with mean variation (see Figure below, top-40 features shown).

In summary, we see similar features predictive of expression variation and single cell noise - although performance is worse for the latter. Despite these encouraging results, we would prefer not to add them to our manuscript since these are very preliminary and will require further confirmation on a larger number of studies, and in-depth analysis of single-cell expression noise is non-trivial. We therefore believe it is out of scope for this manuscript.

- *The weaker effects of distal DHS over promoter architecture in the control of transcriptional precision could be partly attributed to the assignment of DHS to the incorrect target genes. To minimize these miss-assignments, the authors could take into account co-occurrence of DHS and target genes within the same TADs, compatibility rules between genes and enhancers as described by the Stark lab, similarity in chromatin states, etc.*

Authors' response: We agree with the reviewer that the enhancer-gene assignments in our study (i.e. distal DHS sites within 10 kb of the TSS) are not perfect. Defining correct enhancer-to-gene assignments is a current huge challenge in genomics and still a very active field of research, including in the Furlong lab. As in many studies, we have simply assigned DHS to genes based on distance. Alternatively, we could use published lists of annotated enhancers assigned to target genes, which are partial and may create their own selection biases (e.g. genes that are studied more often will have more enhancers assigned) and will only comprise enhancers for a subset of the genes from our dataset.

Nevertheless, to address the reviewer's comment, in an effort to minimize mis-assignments, we have tested the predictions of our model using two sets of enhancer-to-gene annotations, and can report that these tests confirmed our overall results that enhancer complexity is correlated with lower expression variation.

The two additional, orthogonal sets of gene-to-enhancer assignments are:

1. Reporter assays (Vienna Tile enhancers, Kvon et al 2014) – 783 enhancers assigned to 308 genes (143 genes from our set)
2. Machine learning approach McEnhancer (Hafez et al 2017) – 13178 enhancers assigned to 1621 genes (902 genes from our set)

For these two subsets of genes, we calculated the number of enhancers per gene, which was moderately correlated (Spearman rho of 0.35 and 0.31) with the number of distal DHSs (our proxy for number of enhancers). Next, we computed correlation between the number of enhancers and gene expression variation (see table below). Similar to our reported results we found a negative correlation with expression variation for narrow promoter genes (rho=-0.22 in our set vs -0.25 and -0.21). For broad genes, we found a weak positive correlation in our naive linking (0.08), while the two sets above gave weak negative effects (-0.03 and -0.11).

	Kvon et al 2014		Hafez et al 2017		Our results	
	rho	N genes	rho	N genes	rho	N genes
broad	-0.03	25	-0.11	239	0.08	1798
narrow	-0.25	118	-0.21	663	-0.22	2276

Importantly for the discussion here, independent of the enhancer-to-gene assignments that we used, our main conclusion that enhancer complexity correlates with reduced expression variation in narrow promoter genes still holds. However, we might be underestimating enhancer effects for broad promoter genes, in particular, because broad promoter genes often cluster in the loci with high gene density which makes enhancer assignment by proximity even more complicated. However, the contra argument is that regulatory elements of broad promoter genes (e.g. house-keeping) are typically located very close to the gene's promoter (Zabidi et al. 2015) and are therefore usually accurately assigned to the correct target gene based on proximity alone.

As the more curated gene-to-enhancer lists are much more limited in terms of the genes for which we can obtain a mapped enhancer at all, we feel our naive mapping is the better of two imperfect solutions. Therefore, instead of adding the additional analysis to the manuscript, we now point out the difficulties and incompleteness of enhancer-to-gene mappings in the Discussion.

Authors' action: We have added the following paragraph to page 31 lines 661-666 of the Discussion:

“In addition, the weaker (though significant) correlation of enhancer complexity with expression variation might be the result of incorrect enhancer to target gene assignment, which is a current huge problem in genomics. While beyond the scope of this study, a systematic analysis of post-transcriptional regulation and enhancer complexity on expression variation would be interesting directions for future research, when such datasets become available.”

- Throughout the manuscript there are many comparisons between the three major groups of genes with different promoter types. Many of these comparisons are presented as boxplots together with *p*-values that measure whether the differences between groups are statistically significant. I think that these plots should also include "Effect size" measurements in order to better illustrate the magnitude of the differences between groups. Given the large datasets being compared, significant *p*-values are probably "easy" to obtain, which does not necessarily mean that the observed differences are neither large in magnitude nor in biological relevance.

Authors' response: Thank you for this suggestion, adding the effect sizes is a very good idea. We have added Cohen's *d* (defined as difference between two means divided by the pooled standard deviation) to most of the boxplot analyses.

Authors' action: We have added Cohen's *d* to the legends of Figures 2b-e, 4a-b, 5a-c,g-h, 6c,e,f and Appendix Figures S3c, S4a-f, S5c-f,g-j, S8c.

- The link between expression variation and differential expression is interesting but should be more thoroughly investigated, since only differential expression in response to environmental stress has been considered. It is known that transcriptional responses to different environmental insults share a significant fraction of genes. Many of these shared "stress response" genes are up-regulated regardless of the insult: is this also true for differentially expressed genes displaying high expression variation?. In other words, is expression variation preferentially associated with gene activation or also with silencing?. On the other hand, the authors should also consider other differential gene expression datasets in which changes in signalling conditions (e.g. treatment with morphogen agonists or antagonists) or genetic perturbations (e.g. KO for a transcription factor) are used instead. Can the authors still observe a clear link between differential expression and expression variation?

Authors' response: We thank the reviewer for these suggestions. We agree that the analysis of the link between expression variation and differential expression could be expanded further. We had more exhaustively evaluated the link in humans based on DE prior, which is not limited to stress response data. We have now added a more extensive analysis on the link between differential expression and expression variation also for *Drosophila*.

To address the first comment, whether differentially expressed genes with high expression variation are mainly up-regulated upon stress, similarly to many shared stress response genes, we have separated genes based on whether they are up- or down-regulated upon stress in the dataset from (Moskalev et al 2015). The majority of genes were down-regulated, and we didn't observe strong differences in expression variation for genes that went up or down upon stress (see figure below, also added to Appendix Fig. S5, 'down-up' means that genes were up-regulated in some stress conditions and down-regulated in the other). So, the answer is no - there seems to be no preferential association with either up- or down-regulated genes.

To address the suggestion of extending our data to include not only stress-responses but also genetic perturbation studies, we have now collected data from 53 differential expression studies with genetic perturbations in *Drosophila* (three datasets from the Furlong lab examining loss-of-function mutants for TFs involved in mesoderm formation and 50 datasets from Expression Atlas). Information about the collected datasets is available as Dataset EV8. In total, there are 136 differential expression datasets for *Drosophila* deposited in Expression Atlas. We used the following criteria to include data from this resource:

- Consider studies where ‘Experimental factor’ field includes keywords ‘genotype’, ‘phenotype’, ‘genetic modification’ or ‘RNAi’ as ‘genetic perturbation’.
- Use absolute log₂-fold change of 1 and FDR of 5% to define differentially expressed genes (default from Expression Atlas website)
- Only include datasets with at least 300 differentially expressed genes defined as specified above
- We excluded one dataset with more than 8000 differentially expressed genes out of ~12 thousand quantified (E-GEOD-28728) and two datasets where significantly differentially expressed genes come from comparisons with outlier samples (E-MEXP-1179 and E-GEOD-3069).

For each study, we took all differentially expressed (DE) genes (union of conditions, if multiple conditions were tested) and compared variation of DE vs. non-DE genes in our dataset (4074 genes in total) using Cohen’s d as the measure of effect size and p-value from Wilcoxon rank test (see Figure below, added as panel to Fig 5). For the majority of studies (83% (44/53)), DE genes were more variable compared to non-DE genes (Cohen’s $d > 0.2$; based on empirical threshold for low effect size proposed in (Cohen 1988)). This strongly agrees with the conclusions drawn from our results for stress response experiments and indicates that expression variation is strongly linked to differential expression in a wide range of very diverse conditions.

Interestingly, we observed a clear pattern between the number of studies that found a gene differentially expressed and expression variation in our data: genes that were rarely found differentially expressed were on average less variable than those observed more often (see figure below - added to Fig 5). One interpretation of this is that genes that are observed only in a specific experiment may be more direct targets of the perturbation, and thus potentially more interesting to follow-up than those that frequently change their expression regardless of treatment. These frequent changers may be genes that are very responsive to any environmental differences between test and control samples or stress induced by the genetic perturbation or additional treatments (some studies included additional experimental factors such as diet or growth condition, see Dataset EV8, field 'factor').

In line with this, we found that the majority of genes that are differentially expressed in multiple studies had narrow promoters (see figure below).

And consistent with our interpretation of the stress-response results, we found that within the narrow promoter group, genes that were differentially expressed in more than 10 studies had significantly lower regulatory complexity, as indicated by various regulatory features (see Fig below - added to Fig. 5 and Appendix Fig. S5).

Finally, the model for predicting expression variation (top-30% vs. bottom-30% variable genes) can also identify genes differentially expressed in multiple experiments (DE in more than 10 datasets vs. DE 0-10 datasets) following the same methodology as described in the manuscript. The model performed with AUC of 0.78 on all genes (mostly, classifying broad vs. narrow promoter genes) and AUC of 0.74 on narrow promoter genes only (see figure below - added to Fig 5):

In summary, we observed similar patterns upon naturally occurring stress conditions and experimentally induced genetic manipulations. The latter might result from the fact that genetic perturbation itself introduces stress to the organism and changes cellular environment hence triggering non-specific response from more variable genes.

Author's action: We included description of the results summarized above in Results (page 22 line 468; pages 23-24 lines 491-519), Discussion (page 33 lines 710-714) and Methods (pages 55-56 lines 1256-1277). Panels mentioned above were added as Appendix Fig. S5a (up- and down-regulated genes), Fig. 5e-l, and Appendix Fig. S5g-j (genetic perturbation experiments).

- Finally, the manuscript would also improve if some of their major findings and claims are experimentally validated. For example, the authors could use CRISPR technology to modify the promoter of some candidate genes and evaluate whether expression variation is affected as predicted by the authors. Although this might not be essential for publication, it would increase the impact of the authors' claims.

Authors' response: We agree with the reviewer that this would be great to have, however such experiments are time-consuming (3-4 months in *Drosophila* if all goes well) and we believe that they are outside the scope of our manuscript, especially during this lock-down phase where our lab has been closed

4th Jun 2020

Manuscript Number: MSB-20-9539R

Title: Predictive features of gene expression variation reveal mechanistic link with differential expression

Thank you for sending us your revised manuscript. We think that the performed revisions satisfactorily address the issues raised by the reviewers. As such, I am glad to inform you that your manuscript is now suitable for publication, pending some minor editorial issues listed below.

Corresponding Author Name: Judith Zaugg, Eileen Furlong

Manuscript Number: MSB-20-9539